# ATTRICI v1.1 - counterfactual climate for impact attribution

Matthias Mengel[1], Simon Treu[1], Stefan Lange[1], Katja Frieler[1]

[1]Potsdam Institute for Climate Impact Research (PIK), Member of the Leibniz Association, P.O. Box 60 12 03D-14412 Potsdam, Germany

Correspondence to: Matthias Mengel (matthias.mengel@pik-potsdam.de)

**Abstract.**

Attribution in its general definition aims to quantify drivers of change in a system. According to IPCC WGII a change in a natural, human or managed system is attributed to climate change by quantifying the difference between the observed state of the system and a counterfactual baseline that characterizes the system's behavior in the absence of climate change, where

"climate change refers to any long-term trend in climate, irrespective of its cause". Impact attribution following this definition remains a challenge because the counterfactual baseline, which characterizes the system behaviour in the hypothetical absence of climate change, cannot be observed. Process-based and empirical impact models can fill this gap as they allow to simulate the counterfactual climate impact baseline. In those simulations, the models are forced by observed direct (human) drivers such as land use changes, changes in water or agricultural management but a counterfactual climate

without long-term changes. We here present ATTRICI (ATTRIbuting Climate Impacts), an approach to construct the required counterfactual stationary climate data from observational (factual) climate data. Our method identifies the long-term shifts in the considered daily climate variables that are correlated to global mean temperature change assuming a smooth annual cycle of the associated scaling coefficients for each day of the year. The produced counterfactual climate datasets are used as forcing data within the impact attribution set-up of the Inter-Sectoral Impact Model Intercomparison Project

(ISIMIP3a). Our method preserves the internal variability of the observed data in the sense that factual and counterfactual data for a given day have the same rank in their respective statistical distributions. The associated impact model simulations allow for quantifying the contribution of climate change to observed long-term changes in impact indicators and for quantifying the contribution of the observed trend in climate to the magnitude of individual impact events. Attribution of climate impacts to anthropogenic forcing would need an additional step separating anthropogenic climate forcing from other

sources of climate trends, which is not covered by our method.

## 1 Introduction

Global mean temperature (GMT) has recently surpassed 1°C warming above pre-industrial levels (IPCC 2018). The impact of the realized change in climate has also started to become detectable in natural, human or managed systems such as freshwater resources, terrestrial water systems, coastal systems, oceans, food production systems, the economy, human health, security and livelihoods (IPCC 2014). The causal chain from climate change to climate impacts is often complex and intertwined with additional drivers, such as changes in management that alter climate-induced changes in crop yields (Butler, Mueller, and Huybers 2018; Iizumi et al. 2018; Zhu et al. 2019) and land-use changes adding to climate-driven changes in biodiversity (Hof et al. 2018).

Attribution in its most general definition aims to quantify the drivers of change in a system. The systems and drivers considered in attribution studies vary between disciplines. In climate science, the 'classical' attribution framework refers to the attribution of changes in the climate system to anthropogenic forcing (Hegerl et al. 2010; WGI contribution to IPCC 2013) ('climate attribution', see first arrow in Fig. 1). It addresses the question: What is the contribution of anthropogenic emissions of greenhouse gases and aerosols or land use changes to observed changes in climatic variables, most prominently temperature and precipitation? As the response of the climate system to these forcings is often veiled by the chaotic nature of the climate system, climate attribution usually builds on probabilistic approaches comparing an entire ensemble of climate model simulations including anthropogenic forcings against a counterfactual ensemble excluding these forcings as e.g. generated within DAMIP (Gillett et al. 2016) to separate forced changes from internal variability. Climate attribution can refer to observed long-term trends (WGI contribution to IPCC 2013, chap. 10) or individual events (Trenberth, Fasullo, and Shepherd 2015; NAS 2016; Stott et al. 2016). Given the probabilistic setting, results are often formulated as statements such that 'Anthropogenic climate forcing has increased the probability of occurrence of the observed trend or the intensity or duration of a specific extreme event'. In a non-probabilistic framework the intensity of an observed event can be attributed to the observed realisation of climate change by comparing the event magnitude in the observed time series to the magnitude of the same event in a detrended version of the observed time series (quantification of the 'contribution of the observed trend to event magnitude', (Diffenbaugh et al. 2017)). This type of attribution to climate change does not address the reasons of the observed climate trend.

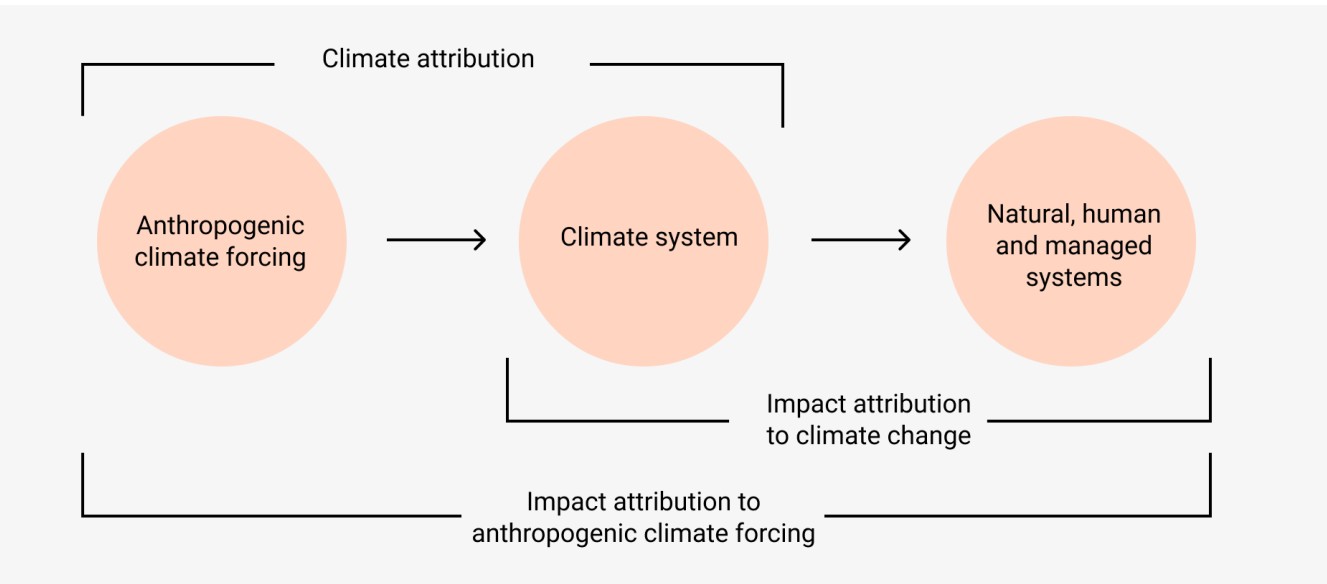

*Figure 1*. *Differences between drivers and affected systems in attribution research. Climate attribution (first arrow) is a focus of IPCC WGI (IPCC 2013) and (climate) impact attribution is a focus of IPCC WGII (IPCC 2014, chap. 18). The methodology and data presented here facilitates the use of (process-based) impact models to attribute observed changes in human, managed and natural systems to climate change (second arrow). The additional step of attribution to anthropogenic climate forcing (first and second arrow) is not addressed here.*

In addition to 'climate attribution', research on 'impact attribution' addresses the question: To what degree are observed changes in natural, human and managed systems induced by observed changes in climate (Fig. 1, second arrow). In the WGII contribution to the IPCC-AR5, an entire chapter was dedicated to the topic including the following definition: An impact of climate change is 'detected' if the observed state of the system differs from a counterfactual baseline that characterizes the system's behavior in the absence of changes in climate (IPCC 2014, chap. 18.2.1) and 'attribution' is the quantification of the contribution of climate change to the observed change in the natural, human or managed system. In both cases "climate change refers to any long-term trend in climate, irrespective of its cause" (IPCC 2014, chap. 18.2.1).

While in principle changes in natural, human and managed systems could also be attributed to anthropogenic climate forcing ('impact attribution to anthropogenic climate forcing', first and second arrow in Fig. 1, (Pall et al. 2011; Schaller et al. 2016; D. Mitchell et al. 2016)), we focus on 'impact attribution to climate change' as described in the WGII definition and introduce a climate dataset that can be used as input to climate impact models to characterize the system's behavior in the absence of climate change (second arrow in Fig. 1). The dataset is derived from the observed realization of climate, excluding the analysis how climate variability could produce alternative realizations of factual or counterfactual climate. The attribution approach is thus deterministic and not probabilistic, focusing on the separation of climate change from direct human influences as potential drivers of changes in the impacted systems. Concerning the internal variability within impacted systems, impact models to date largely do not resolve such variability and model a deterministic response to external drivers. Our approach would however allow for probabilistic attribution to climate change once impact models resolve internal variability.

The method proposed here is designed to generate a stationary climate without long-term changes. The statistical model used to produce this counterfactual climate removes the long-term change correlated with (but not necessarily caused by) large scale climate change, represented by GMT change instead of a simple temporal trend (see Methods). The method preserves

the internal variability of the observed time series by additively (e.g. for temperature) or multiplicatively (e.g. for precipitation) removing a long-term trend, such that a particularly warm or dry day compared to the long-term trend remains a particularly warm or dry day in the counterfactual climate. In this regard, the approach is similar to the subtraction of a climate trend done by (Diffenbaugh and Burke 2019) to attribute historical changes in economic growth orto attribute changes in land area burned by wildfires (Abatzoglou and Williams 2016). However, while both studies subtract the anthropogenic warming derived from climate model simulations, we subtract the realised long-term trend of the data irrespective of its cause (see WGII definition).

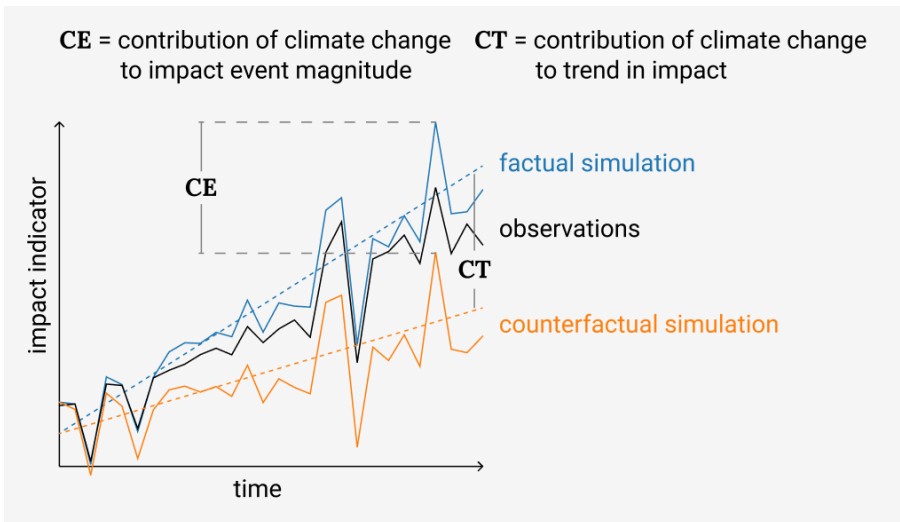

*Figure 2. Attribution of impact event magnitude and trends in impact indicators to trend in climate. First, in an evaluation step it has to be demonstrated that historical impact observations (black line) can be explained by the process-understanding as represented in the applied impact model and available knowledge about historical climate and socio-economic forcings. To this end, the factual simulations forced by historical climate and socio-economic forcings (solid blue line) are compared to the impact observations (solid black line). Secondly, in the attribution step the impact model is driven by counterfactual climate while all other drivers are kept equal to the factual simulation (counterfactual simulation; solid orange line). A comparison between factual and counterfactual simulations allows for the attribution of long term changes (e.g. trends) in the impact indicator to trends in climate (contribution to trend, **CT**). In addition, the contribution of climate change to the magnitude of individual events (impact event attribution) can be determined as the difference between the simulated factual event magnitude and the counterfactual impact event magnitude (**CE**).*

The stationary climate dataset can then be used as input to climate impact models for impact attribution to climate change, as illustrated in Fig. 2. In a first step the climate impact model forced by observed climate and socio-economic drivers has to demonstrate to be able to reproduce the observed changes in natural, human and managed systems as measured by an impact indicator (comparison of black and blue solid lines in Figure 2). The attribution of the observed changes in natural, human and managed systems is built on a high explanatory power of the factual simulations. Then, in a second step, that factual simulation can be compared to a counterfactual simulation, forced by counterfactual climate but otherwise the same input as in the factual simulation. Such a comparison allows for a quantification of the contribution of climate change to both the observed trend in the impact indicator (CT in Fig. 2) and the observed magnitude of an individual impact event (CE in Fig. 2). This assumes that the climate impact model calibrated to perform well in the factual simulation performs robustly also with counterfactual climate input data.

Process-based impact models such as those taking part in the ISIMIP project ([www.isimip.org](www.isimip.org)) are ideal tools to address impact attribution as they generally describe the response of natural, human or managed systems not only to climate but also direct (human) drivers. For example, crop models can simulate the response of crop yields to changes in land use, irrigation patterns, fertilizer input and crop varieties (Lobell, Schlenker, and Costa-Roberts 2011; Challinor et al. 2014; Minoli et al. 2019). Similarly, hydrological models can be used to simulate how dam construction and water withdrawal affect river discharge (Veldkamp et al. 2017, 2018). In addition, those models allow for a process-based representation of the extent of e.g. river floods and droughts that can be combined with maps of asset distribution and empirical damage functions to estimate the direct economic damages induced by weather extremes. The impact attribution framework could then be used to approximate the contribution of climate change to observed trends in reported damages. Using process-based climate impact models, this contribution can be explicitly separated from changes in damages driven by changes in exposure or vulnerability. In this regard it goes beyond available approaches of damage attribution that attribute to anthropogenic climate forcing but simply estimate the contribution of anthropogenic climate forcing to observed damages by multiplying the fraction of attributable risk associated with weather extremes by the observed damage (Frame et al., 2020). In the same way, it could improve the attribution of health impacts (D. Mitchell et al. 2016).

In this paper we introduce a detrending method tailored to support impact attribution and illustrate its application to one of the observational climate datasets provided within ISIMIP3a ([https://protocol.isimip.org/protocol/ISIMIP3a](https://protocol.isimip.org/protocol/ISIMIP3a), see data section below). The quality of the associated impact attribution studies will critically depend on the quality of that observational dataset. Deficits in the observational data may lead to artefacts in the derived historical trends. For the dataset used here, we identify some of those artifacts. Since it is expected that other artifacts will be found for other observational datasets, impact attribution studies should ideally be based on a range of different observational datasets to facilitate a quantification of the contribution of observational climate data uncertainty to the uncertainty of the attribution results. This is also planned within ISIMIP3a. For the dataset considered here and potential additional ones we propose a collection of control plots that should be used to scan the observational climate data for artifacts in preparation of each individual impact attribution study. While we provide the control plots for a set of large-scale world regions and all climate variables covered by our observational climate dataset, they should be adjusted to the regions and variables of interest in an impact attribution study as part of the analysis of the factual impact simulation (Fig. 2). Low-quality factual climate forcing data is expected to result in a low-quality reproduction of observed variations in the impact indicators of interest. If that is the case, the simulation set-up outlined here does not allow for an attribution of the observed changes in impacts to climate change.

## 2 Data

For ISIMIP3, we construct counterfactual climate data for the global observational dataset GSWP3-W5E5. This dataset has daily temporal and 0.5° spatial resolution and consists of two parts, W5E5 v2.0 for the period 1979-2019 and GSWP3 v1.09 homogenized with W5E5 for the period 1901-1978. In the following, we describe these two parts as well as why and how they were combined for ISIMIP3.

The GSWP3 v1.09 dataset is from the third phase of the Global Soil Wetness Project (GSWP3), an ongoing land model intercomparison project, which shares its experiment protocol with "land-hist" of the Land Surface, Snow and Soil moisture Model Intercomparison Project (LS3MIP; van den Hurk et al. 2016) and covers the years 1901-2014 (Kim 2017). It is a dynamically downscaled and bias-adjusted version of the 20th Century Reanalysis (20CR; Compo et al. 2011) and has been used as a meteorological forcing dataset in several climate impact assessments such as those carried out in ISIMIP2a (e.g.,

(Müller Schmied et al. 2016; Chang et al. 2017; Schewe et al. 2019; Padrón et al. 2020)) as well as in broader modeling studies (e.g., Krinner et al. 2018; Tangdamrongsub et al. 2018; Tokuda et al. 2019). GSWP3 is also provided for the impact model evaluation task within ISIMIP3a.

20CR assimilates subdaily surface pressure and sea-level pressure observations and uses monthly sea-surface temperature (SST) and sea-ice distributions from the Hadley Centre Sea Ice and SST dataset (HadISST; Rayner et al. 2003) as lower boundary conditions. To produce GSWP3, the first of the 56 members of the 20CR ensemble was dynamically downscaled to T238 (about 0.5°) spatial resolution using the incremental correction of a single member (ICS) method of (Yoshimura and Kanamitsu 2013) and the Scripps Institution of Oceanography (SIO)/Experimental Climate Prediction Center (ECPC) Global Spectral Model (GSM) with spectral nudging (Yoshimura and Kanamitsu 2008) and vertically weighted damping coefficients (Hong and Chang 2012). The ICS method additively adjusts the prognostic fields of a single ensemble member such that at the monthly time scale each adjusted field is identical to the corresponding ensemble mean field while all higher-frequency parts of the fields are retained. Hence, the adjusted fields represent the 20CR best estimates at the monthly time scale while they do not suffer from the increase of synoptic variability over time found in the 20CR ensemble mean (Compo et al., 2011) that is due to a decrease of the ensemble spread over time, which in turn reflects the increase in available observational constraints (Yoshimura and Kanamitsu, 2013).

The downscaled 3-hourly data were then bilinearly interpolated from T238 to a regular 0.5° latitude-longitude grid. In addition, selected variables (precipitation, surface downwelling shortwave and longwave radiation, near-surface wind speed, near-surface air temperature, surface air pressure and near-surface specific humidity) were bias-adjusted with different methods and observational reference datasets. Precipitation was bias-adjusted at the monthly time scale using an undercatch-corrected version of the Global Precipitation Climatology Centre (GPCC) Full Data Monthly Product Version 7 (Schneider et al. 2014). The bias adjustment was done by rescaling all monthly mean values to the GPCC estimates. Radiation was bias-adjusted at the daily time scale using Surface Radiation Budget (SRB; Stackhouse et al. 2011) primary-algorithm estimates of daily mean values from SRB release 3.1 for longwave radiation and SRB release 3.0 for shortwave radiation. Since those estimates are only available for 1983-2007, bias-adjusted daily values were computed as the sum of a rescaled monthly mean value and a rescaled daily anomaly from the monthly mean, with the rescaling done such that, for the 1983-2007 time period, both the monthly mean climatology and the anomaly standard-deviation climatology matched the respective SRB estimates. Wind speed was bias-adjusted at the monthly time scale over land using mean monthly climatologies from the Climatic Research Unit (CRU) CL2.0 dataset (New et al. 2002). The bias adjustment was done by monthly rescaling such that the 1961-1990 mean monthly climatology matched that of CRU CL2.0. Temperature, pressure and humidity were bias-adjusted at the 3-hourly time scale using the WATCH forcing data methodology (Weedon et al. 2014) and monthly mean temperatures plus monthly mean diurnal temperature ranges from the CRU TS3.23 dataset (Harris, Jones, and Osborn 2014), which covers the full 1901-2014 time period.

As a consequence of its derivation, the quality of the GSWP3 data varies over time. It varies in line with variations in the availability of the pressure, SST and sea-ice observations used to produce 20CR (Compo et al. 2011; Rayner et al. 2003) as well as with variations in the availability of the precipitation and temperature observations used to bias-adjust GSWP3 (Schneider et al., 2014; Weedon et al., 2014). Examples of temporal inhomogeneities in GSWP3 that are relevant for this study include artificial drying trends over northwest China and the Tibetan Plateau over the first half of the 20th century (Fig. 10) that are inherited from GPCC (Chen and Frauenfeld 2014), and spurious trends in shortwave radiation and wind speed over Alaska, Northern Canada and Greenland over the first half of the 20th century (Figs. 9 and S12), which are

related to artificial extratropical cyclone trends in 20CR over that time period (Wang et al. 2013). Generally, the quality of 20CR, and hence GSWP3, becomes relatively stable around mid-century over the Northern Hemisphere, earlier over Europe and later over the Southern Hemisphere, in line with variations in the availability of pressure observations for data assimilation in the reanalysis (Compo et al., 2011).

The W5E5 v2.0 dataset (Lange 2019a) was compiled to support the bias adjustment of climate input data carried out within ISIMIP3b and covers the years 1979-2019. It combines the WFDE5 v2.0 dataset (WATCH Forcing Data methodology applied to ERA5 reanalysis data; (Cucchi et al. 2020)) over land with data from the latest version of the European Reanalysis (ERA5; Hersbach et al. 2020) over the ocean. WFDE5 is a meteorological forcing dataset based on ERA5. For the variables included, it is a spatially aggregated (to 0.5°) and bias-adjusted version of ERA5. Compared to 20CR used for GSWP3, many more observations were used for data assimilation in ERA5, including precipitation observations (Hersbach et al. 2020). That is why we consider ERA5 to better represent reality than 20CR for 1979 onwards. Similarly, WFDE5 is considered to better represent reality than GSWP3, in particular with respect to day-to-day variability for variables that were bias-adjusted using only monthly mean values in both datasets, such as temperature and precipitation.

Since W5E5 is considered the more realistic dataset but only covers 1979–2019 it was extended backward in time to generate GSWP3-W5E5 for ISIMIP3. In this extended dataset, GSWP3 data for 1901–1978 were homogenized with W5E5 data using the ISIMIP2BASD v2.5 quantile mapping method (Lange 2019b, 2020). The resulting GSWP3-W5E5 data are identical to the original W5E5 data from 1979 onwards but different from the original GSWP3 data before 1979. The goal of the homogenization was to smooth the transition from one dataset to the other in 1978/1979. To that end, for every climate variable and grid cell individually, the original GSWP3 time series for 1901–2004 were quantile-mapped to time series which have the same trends but whose distributions match those of the corresponding W5E5 data over the 1979–2004 reference period. The resulting, homogenized GSWP3 data for 1901–1978 were then used to extend W5E5 backward in time. The preservation of trends implies that differences between trends in GSWP3 and W5E5 data were not homogenized. Consequently, some inhomogeneities at the 1978/1979 transition remain. This problem particularly affects surface downwelling shortwave radiation over Northern Europe and the Mediterranean Basin (Fig. 8) as discussed further in the results section.

## 3 Methodology

Assuming that "climate change refers to any long-term trend in climate, irrespective of its cause" (IPCC 2014, chap. 18) we here present a method to generate time series of stationary climate data from observational daily data by removing the long-term trend while preserving the internal day-to-day variability. In the following, we first describe the general characteristics of our approach followed by a more detailed formal description of the method. Then we introduce the set of global and regional evaluation plots we recommend to regionally adjust and consider within each attribution study using the counterfactual data generated here or when applying the detrending approach to other observational climate data.

### 3.1 Detrending method

A very basic detrending approach would fit a linear temporal trend for all data of each day of the year assuming normally distributed residuals and remove the estimated trends from the data for each day of the year separately. In this approach the trend estimates would not only vary according to systematic variations in trends from one day of the year to the other but also randomly fluctuate from one day of the year to the next one in terms of the uncertainties associated with the individual

estimates.

We go beyond this very basic approach by i) using global mean temperature change instead of time as a potentially powerful predictor of regional changes in climate, ii) allowing for non-normal distributions of the unexplained random year-to-year fluctuations of data per day of the year, and iii) ensuring a smooth variation of estimated model parameters from one day of

the year to the other.

The use of GMT change, $T$, as the predictor of regional climate change is motivated by the classical pattern scaling approach (Santer et al. 1990; T. D. Mitchell 2003), with newer approaches including additional predictors such as a distinction between land and sea to improve accuracy (Herger and Sanderson 2015). Here, $T$ is GMT change since 1901

smoothed by singular spectrum analysis (SSA, Ghil et al. 2002) with a smoothing window of 10 years (Fig. 3). The smoothing of the predictor is applied because we only want to remove long-term trends from the regional climate time series. Natural climate variability on shorter time scales due to phenomena such as the El Niño–Southern Oscillation is retained.

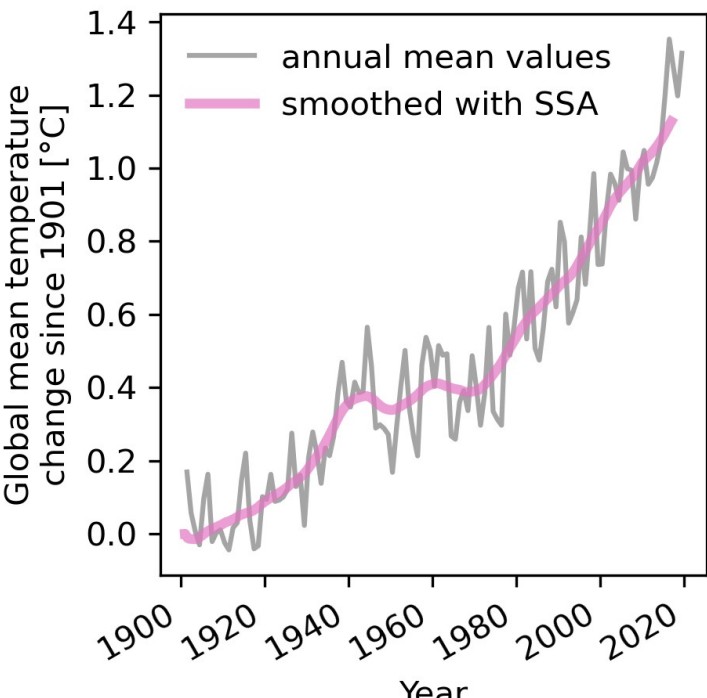

*Figure 3. Time series of GMT change since 1901 derived from GSWP3-W5E5 near-surface air temperature data. Shown are annual mean GMT change (grey) and GMT change smoothed by SSA with a smoothing window of 10 years (pink). The smoothed GMT change is used as the predictor of regional climate change in our detrending model (denoted by $T$ in the text).*

Using $T$ as the predictor means that we remove long-term trends in regional climate to the extent that those are correlated with GMT change, but irrespective of the cause of global warming. The success of the detrending is evaluated by a number of control measures described in Sect. 3.3.

For each day of the year $t$ the detrending is done with quantile mapping (Wood et al. 2004; Cannon, Sobie, and Murdock 2015; Lange 2019b) from the factual distribution $A(T,t)$ to the counterfactual distribution $A(T=0,t)$. The dependence of $A$ on $T$ is modelled via the expected value $\mu$ of the distribution, using a generalized linear model (GLM) or beta regression (Ferrari and Cribari-Neto 2004) with a link function $g$ defined by $g(\mu(T,t))=c_0(t)+c_1(t)T$. The link function $g$ is used to account for climate variables that can only be positive (in that case, $g(x)=\ln(x)$) or can only take values between 0 and 1 (in that case, $g(x)=\ln(x/(1-x))$). In all other cases $g(x)=x$. See Table 1 for an overview of which distributions and link functions are used for the different climate variables, see Sect. 3.2 for further details. For variables modeled by a Gaussian distribution, the variance $\sigma^2$ of $A$ is assumed to stay constant for each day of the year, i.e. $\sigma^2$ does not vary with $T$ but only depends on $t$. For non-Gaussian distributions, the variance is assumed to change with the expected value. In that case we assume the shape of the distribution to stay constant for each day of the year.

We use harmonics for the representation of the annual cycle, i.e. the dependence of the coefficients $c_0(t)$ and $c_1(t)$ on the day of the year $t$. Specifically, we use

$$g(\mu(T,t))=a_0(T)+\sum_{k=1}^{n} a_k(T)\cos(k\omega t)+b_k(T)\sin(k\omega t) \qquad (1)$$

to model the dependence of $\mu$ on $T$ and $t$. Here, $\omega=\dfrac{2\pi}{365.25}$ and $n=4$ Fourier modes are used to model the annual cycle.

The GMT-change dependence of the Fourier coefficients $a_k, b_k$ is modelled linearly,

$$a_k(T)=a_k^{(slope)}T+a_k^{(intercept)}; k=0,1,...,n \qquad (2)$$

and similarly for $b_1, b_2, ..., b_n$.

The distribution parameters that only depend on $t$ are modelled using

$$\ln(v(t))=a_0^{(constant)}+\sum_{k=1}^{n} a_k^{(constant)}\cos(k\omega t)+b_k^{(constant)}\sin(k\omega t), \qquad (3)$$

where $\omega$ and $n$ have the same values as in Eq. (1) and $v$ represents $\sigma$ for the Gaussian distribution, $k$ for the Gamma distribution, $\alpha$ for the Weibull distribution and $\phi$ for the Beta distribution (see Table 1 and Sect. 3.2).

**Table 1:** *Climate variables covered by ISIMIP3a counterfactual climate datasets. Listed are each variable's short name and unit as well as the statistical distribution and link function used for detrending it. Also specified is the dependence of the distribution parameters on GMT change, $T$, and day of the year, $t$, as used in our GLM. The variables tasrange and tasskew are auxiliary variables used to detrend tasmin and tasmax.*

| Variable | Short name | Unit | Statistical distribution | Link function |
|---|---|---|---|---|
| Daily Mean Near-Surface Air Temperature | tas | K | Gaussian with mean value $\mu(T,t)$ and standard deviation $\sigma(t)$ | $g(\mu)=\mu$ |
| Daily Near-Surface Temperature Range | tasrange | K | Gamma with mean value $\mu(T,t)$ and shape $k(t)$ | $g(\mu)=\ln(\mu)$ |
| Daily Near-Surface Temperature Skewness | tasskew | 1 | Gaussian with mean value $\mu(T,t)$ and standard deviation $\sigma(t)$ | $g(\mu)=\mu$ |
| Precipitation | pr | kg m$^{-2}$ s$^{-1}$ | For wet or dry day: Bernoulli with dry day probability $p(T,t)$ | $g(p)=\ln(p/(1-p))$ |
| | | | For intensity of precipitation on wet days: Gamma with mean value $\mu(T,t)$ and shape $k(t)$ | $g(\mu)=\ln(\mu)$ |
| Surface Downwelling Shortwave Radiation | rsds | W m$^{-2}$ | Gaussian with mean value $\mu(T,t)$ and standard deviation $\sigma(t)$ | $g(\mu)=\mu$ |
| Surface Downwelling Longwave Radiation | rlds | W m$^{-2}$ | Gaussian with mean value $\mu(T,t)$ and standard deviation $\sigma(t)$ | $g(\mu)=\mu$ |
| Surface Air Pressure | ps | Pa | Gaussian with mean value $\mu(T,t)$ and standard deviation $\sigma(t)$ | $g(\mu)=\mu$ |
| Near-Surface Wind Speed | sfcwind | m s$^{-1}$ | Weibull with shape $\alpha(t)$ and scale $\beta(T,t)$ | $g(\beta)=\ln(\beta)$ |
| Near-Surface Relative Humidity | hurs | % | Beta with with mean value $\mu(T,t)$ and dispersion $\phi(t)$ | $g(\mu)=\ln(\mu/(1-\mu))$ |
| Near-Surface Specific Humidity | huss | kg kg$^{-1}$ | Derived from hurs, ps and tas | |
| Daily Minimum Near-Surface Air Temperature | tasmin | K | Derived from tas, tasrange and tasskew | |
| Daily Maximum Near-Surface Air Temperature | tasmax | K | Derived from tas, tasrange and tasskew | |

295

By limiting the number of Fourier modes to four we reduce the number of coefficients to be estimated and ensure a smooth variation of the long-term trend in $\mu$ over the course of the year but still capture seasonal to sub-seasonal patterns such as monsoon season onsets. Setting $n=4$ in Eq. (1) leads to a total of eighteen slope and intercept parameters to describe the expected value $\mu$ in terms of $T$ and $t$. Setting $n=4$ in Eq. (3) means that nine parameters are used to describe the dependence of $\sigma, k, \alpha$ and $\phi$ on $t$.

We use a Bayesian approach to estimate all of these parameters. This requires the specification of prior distributions of the model parameters. Similar to regularization techniques in frequentist approaches, the prior allows to focus the model fitting on plausible parameter values. This is particularly important for numeric stability when the logit and logarithm link functions are applied. We use a zero-centered Gaussian prior for all parameters and all climate variables because we normalize the data before parameter estimation. We use a standard deviation of 1.0 for $a_0^{(intercept)}$, a standard deviation of $1/(2k-1)$ for $a_k^{(intercept)}$, $k=1,...,4$, and a standard deviation of 0.1 for $a_k^{(slope)}$, $k=0,...,4$. Our choice of priors for $a_k^{(intercept)}$ is based

on the assumption that the first mode with a period of one year explains the largest part of the annual cycle and higher-order modes have decreasing influence. However this is only a prior assumption, i.e. if the data show different patterns, they can
still be captured by our model. For $a_k^{(constant)}$ we use the same priors as for $a_k^{(intercept)}$. We use the same priors for the parameters $b_k$. We technically implemented the model fitting by use of  the pymc3 python package (Salvatier, Wiecki, and Fonnesbeck 2016). Before the regression, all time series are normalized to simplify the Bayesian model parameter estimation. To restore the original units, the normalization is reversed after detrending.

The overall intention of our approach is to find appropriate parameter values such that $A(T,t)$ captures long-term trends in the variables that can be removed by setting $T$ to zero. This is important because the counterfactual distributions are then defined by $A(T=0,t)$. As an example, the factual $\mu(T,t)$ and the counterfactual $\mu(T=0,t)$ as well as the associated daily values of one particular tas time series are shown in Fig. 4. The difference between the expected values of distribution $A(T,\cdot)$ (black line) and $A(T=0,\cdot)$ (orange line) is due to a vertical shift that is composed of a linear increase with $T$ captured by $a_0$ and a change in the amplitude and phase of the annual cycle captured by the Fourier coefficients $a_k$ and $b_k$, $k>0$. The counterfactual daily data are generated by quantile mapping, i.e. an observed value $x$ that corresponds to a certain quantile of the factual distribution $A(T,t)$ is mapped to the counterfactual value $x'$ that corresponds to the same quantile of the counterfactual distribution $A(T=0,t)$. We illustrate this for an observed value x that corresponds to the 95th percentile of the factual distribution in Fig. 4: We first obtain the cumulative probability of the factual (i.e. observed) temperature (large black dot in panel a) from the factual cumulative distribution function (CDF) (black line in panel b). We then obtain the counterfactual temperature (large orange dot in panel a) from the counterfactual CDF (orange line in panel b).

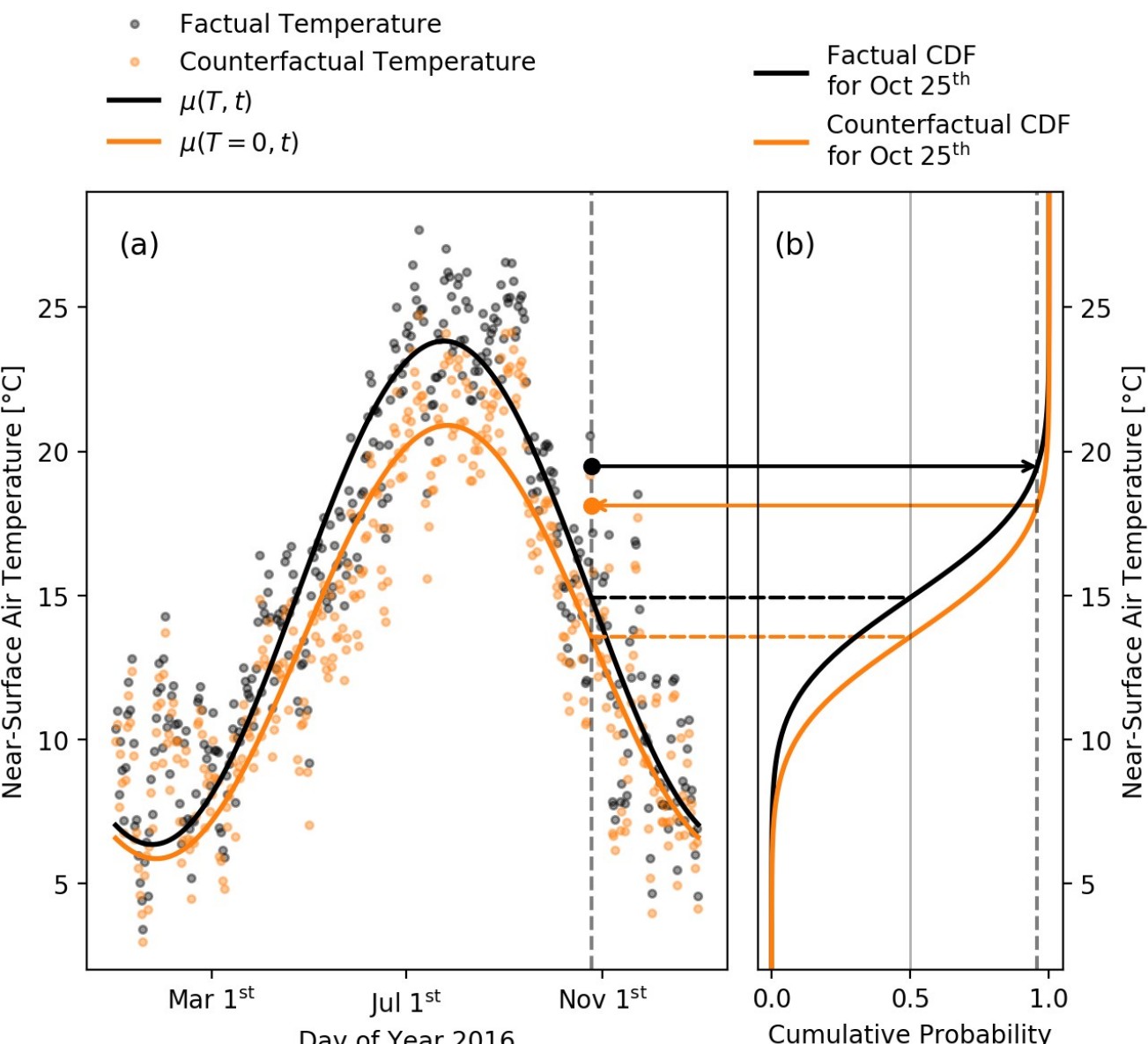

***Figure 4.*** *Illustration of detrending with quantile mapping sensitive to the annual cycle. Panel (a) shows the factual (black*
*points) and counterfactual (orange points) daily mean near-surface air temperature data for the year 2016 of GSWP3-W5E5*
*for a single grid cell in the Mediterranean region at 43.25°N, 5.25°E. In panel (a), the black and orange lines show the*
*temporal evolution of the expected value μ of the factual and the counterfactual distribution. In panel (b), the black and*
*orange lines show the factual and counterfactual cumulative distribution function (CDF) for a single day (October 25[th],*
*2016). The single large black and orange point on the dashed vertical line in panel (a) highlight the factual and*
*counterfactual value on October 25[th]. They correspond to the 95[th] percentile in their respective distributions.*

## 3.2 Model choices for each climate variable

**Near-surface air temperature, surface air pressure, and surface downwelling longwave radiation.** We use the Gaussian
distribution to model these variables as their values are far from their physical lower bound of zero.

**Daily minimum and maximum near-surface air temperature**. They provide a measure of the diurnal temperature cycle in
the daily resolved dataset. We do not estimate counterfactual time series for *tasmin* and *tasmax* directly to avoid large
relative errors in the daily temperature range as pointed out by (Piani et al. 2010). Instead we construct counterfactuals for

the auxiliary variables *tasrange = tasmax - tasmin* and *tasskew = (tas - tasmin) / tasrange* that then determine the tasmin and tasmax counterfactuals (Piani et al. 2010). We use the Gamma distribution to model *tasrange* since it has a lower bound at

zero. The expected value is modeled according to Eq. (1). The skewness of daily near-surface temperature *tasskew* is modeled by a Gaussian distribution. While theoretically bounded, *tasskew* is never close to its bounds of zero and one. This justifies the Gaussian model choice.

**Precipitation.** We use a mixed Bernoulli-gamma distribution (Gudmundsson et al. 2012) for precipitation, i.e. the distribution of wet versus dry days is described by a Bernoulli distribution with *p* describing the probability of dry days

while the intensity of precipitation on wet days is assumed to follow a gamma distribution. A day is considered dry if the amount of precipitation is below 0.1 mm a day. Wet days are all days where the threshold is exceeded. We describe the gamma distribution by its expected value and a shape parameter $k$. We assume that the expected value, $p$, of the Bernoulli distribution and the expected value of the gamma distribution vary with $T$ and $t$ while the shape parameter $k$ of the gamma distribution is assumed to only vary with *t*. If the probability of dry days, $p_{factual}$, of the factual distribution $A(T,t)$ is

larger than the probability of dry days, $p_{counterfactual}$, of the counterfactual distribution $A(T=0,t)$, dry days are turned into wet days at random with probability $p_{factual} - p_{counterfactual}$ by assigning them a small precipitation amount above the wet-day threshold. This random conversion of dry days into wet days may result in physical inconsistencies with other climate variables. These inconsistencies are small by design since the new wet days are the least wet of all counterfactual wet days.

**Surface Downwelling Shortwave Radiation.** Physically bound to positive numbers, the limit is only reached in the special

case of the polar night. We thus use a Gaussian distribution to model *rsds*. If quantile mapping leads to negative values, we use the original value instead.

**Near-surface wind speed.** We use a Weibull distribution to model surface wind speed. The distribution has a shape parameter $\alpha$ and a scale parameter $\beta$, which both need to be positive. The expected value of the Weibull distribution is given by $\beta \Gamma(1+1/\alpha)$ with the Gamma function $\Gamma$. We model the scale parameter $\beta$ by Eq. (1) using the natural logarithm as

the link function. We handle the shape parameter similar to the standard deviation of the Gaussian distribution, being independent of GMT change but varying with *t*.

**Near-surface relative humidity.** Near-surface relative humidity *hurs* is positive and less than or equal to one. We assume *hurs* to follow a beta distribution. Its expected value is allowed to vary with $T$ and $t$. The associated coefficients are estimated using a beta regression model (Ferrari and Cribari-Neto 2004) and Eq. (1) for the expected value while the

dispersion parameter, $\phi$, is assumed to only vary with *t*.

**Near-surface specific humidity.** The counterfactual for *huss* is derived from counterfactual *tas*, *ps* and *hurs* using the equations of (Buck 1981) as described in (Weedon et al. 2010).

### 3.3 Evaluation method

To evaluate the detrending method and the counterfactual GSWP3-W5E5 data we use the difference between multi-year

averages of each climate variable over the beginning of the time period (1901-1930) and multi-year averages over the end of the time period (1990-2019) as a measure of the trend. We compare this trend measure between the observed data and the counterfactual data, for which it should be close to zero (Figs. 5 and 6). In addition, we propose to plot the entire time series for regionally averaged annual (or seasonal) mean values for both the original and the counterfactual climate data. Here, we do so for annual regional averages over 21 world regions (Giorgi and Francisco 2000), see left panels of Figs. 7-10 and

supplementary figures, but propose to adjust the regions and season for each attribution study individually according to its focus. For our specific observational dataset we add annual regional averages of the original GSWP3 data to check if the

homogenisation of GSWP3 with W5E5 has introduced artificial trends in the factual GSWP3-W5E5 data. To evaluate the performance of the detrending method for each day of the year we propose to compare the 1990-2019 regional mean climatology of the counterfactual data to the 1901-1930 regional mean climatology of the factual data for each region of interest (right panels of Figs. 7-10 and supplementary figures).

**4 Results**

The counterfactual dataset evaluated in the following is free to download through the ISIMIP data portal (https://data.isimip.org/search/climate_scenario/counterclim/) along with the underlying original data. Our method strongly reduces the observed difference between multi-year averages over the beginning of the century (1901-1930) and the end of the observational period (1990-2019) for most locations and variables (Figs. 5 and 6). The remaining differences are largest for precipitation over the Tibet region and for wind speed over Greenland. In the following we exemplarily zoom into these regions to resolve the temporal evolution of the regionally averaged factual and counterfactual data (Figs. 7-10, left panels) and evaluate the detrending for each day of the year (Figs. 7-10, right panels). We start with temperature and precipitation in Northern Europe where the detrending works well and then focus on regions where the factual data show artefacts that may make them inadequate for impact attribution within the proposed set-up.

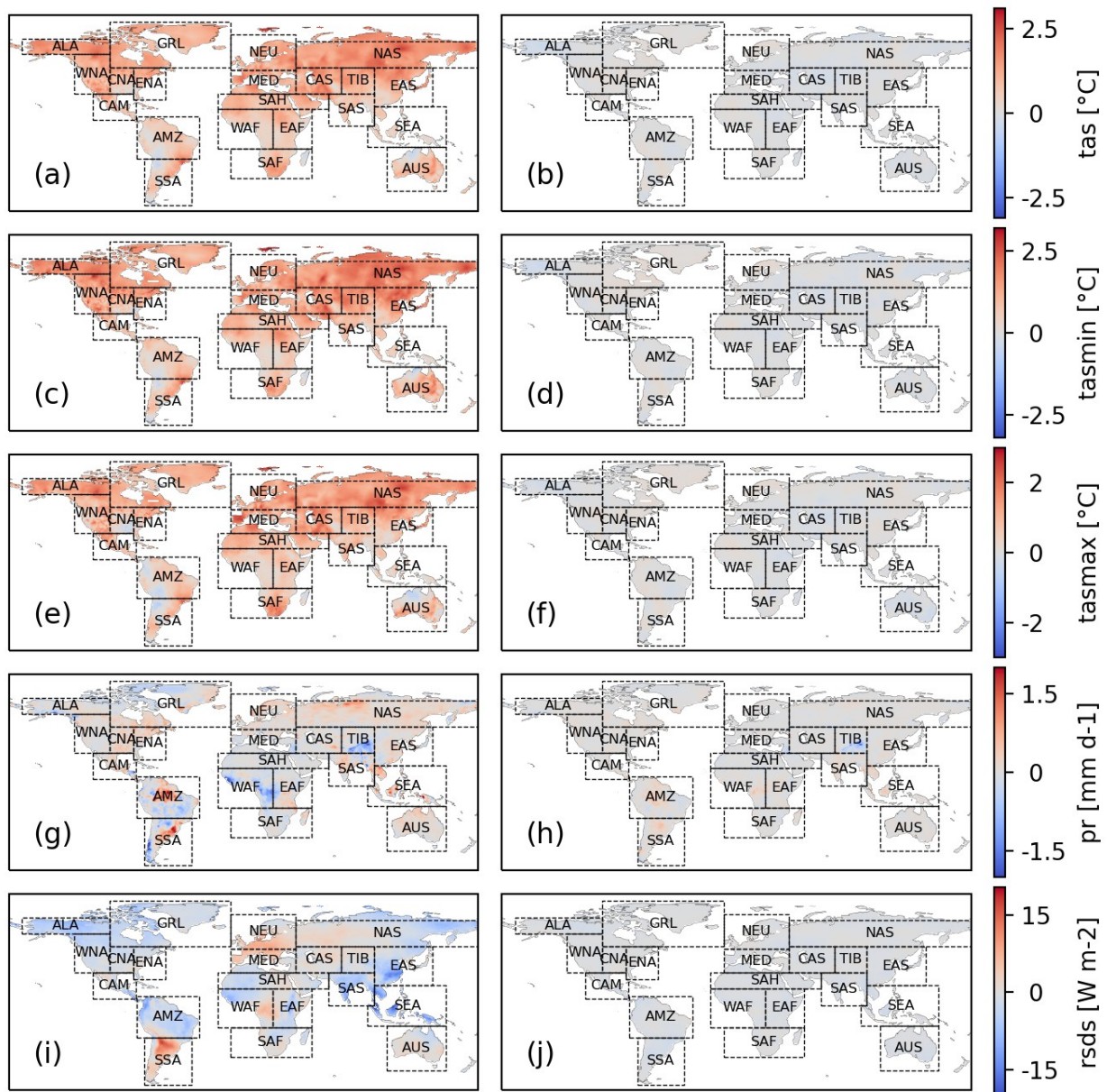

**Figure 5.** *Differences between multi-year averages over the late (1990-2019) and early (1901-1930) time period for the factual (left) and counterfactual (right) GSWP3-W5E5 dataset. Results are shown for tas, tasmin, tasmax, pr and rsds (from top to bottom). Rectangles show the 21 world regions from (Giorgi and Francisco 2000)). Note that the color scale is capped for precipitation, i.e., values below -2 mm d-1 and above 2 mm d-1 are displayed in dark blue and dark red, respectively.*

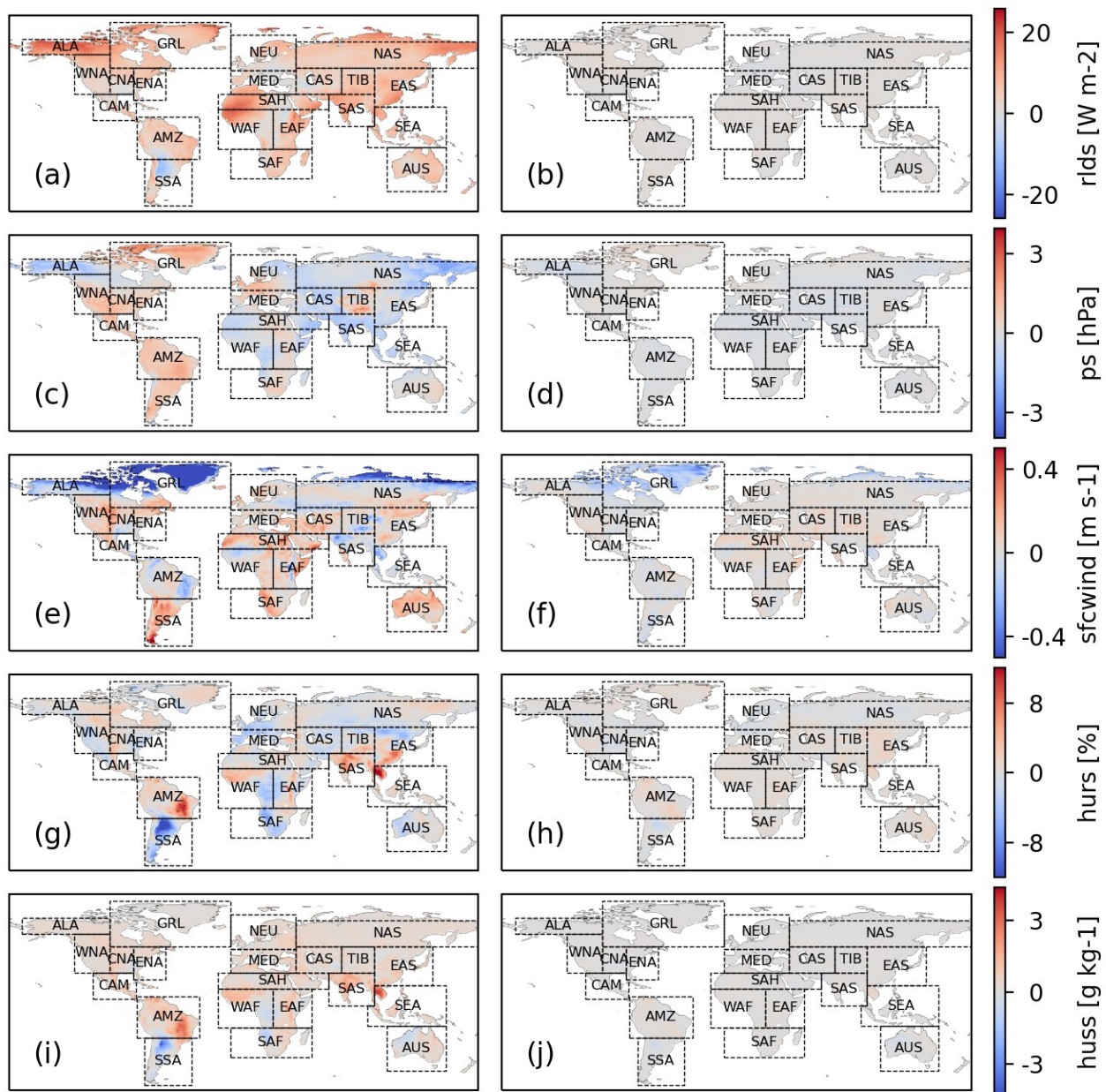

***Figure 6.*** *Same as Fig. 5 but for rlds, ps, sfcwind, hurs and huss. Note that the color scale is capped for wind at -0.5 and 0.5 m s-1 and for hurs at -12 and 12 %. Values below and above those bounds are displayed in dark blue and dark red, respectively.*

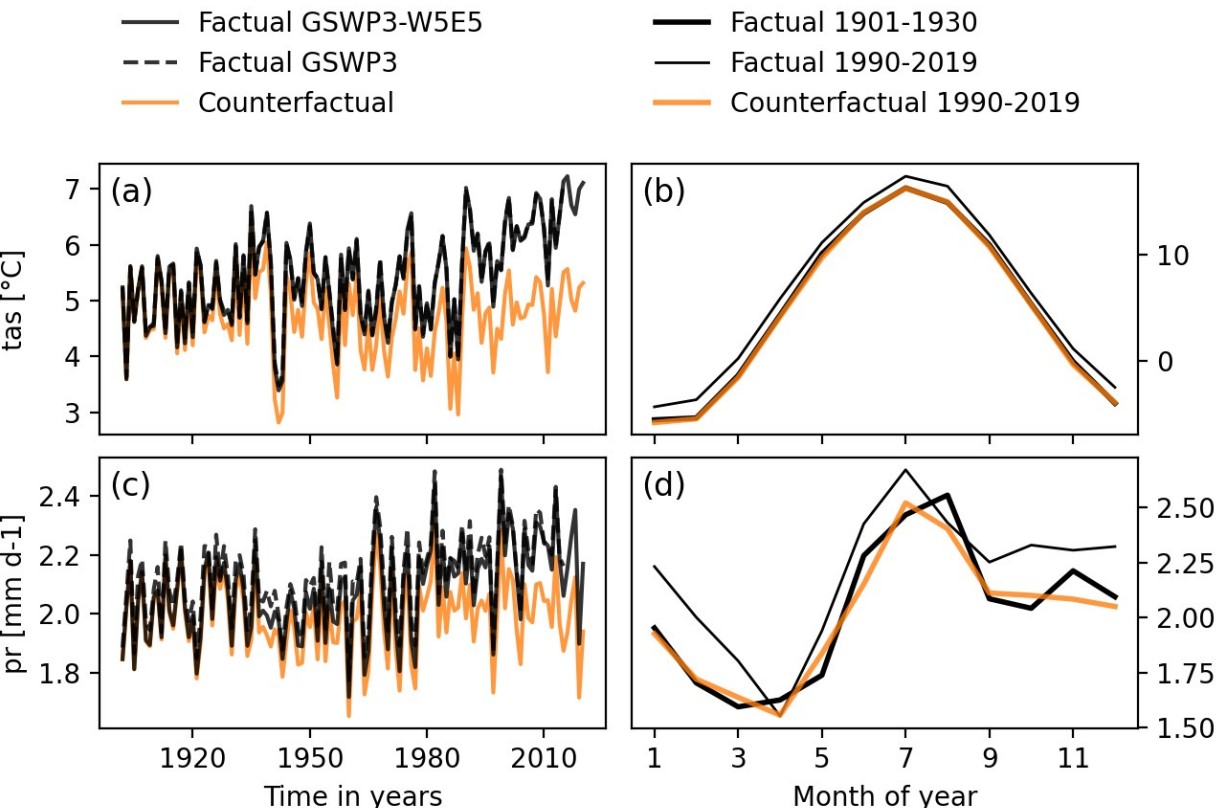

***Figure 7.*** *Panels (a) and (c) show annual regional mean time series of factual GSWP3-W5E5 data (solid black line), factual GWSP3 data (dashed black line) and counterfactual GSWP3-W5E5 data (orange line) for near-surface air temperature (a) and precipitation (c) over Northern Europe (NEU). Panels (b) and (d) show multi-year regional mean climatologies for near-surface air temperature (b) and precipitation (d) of factual and counterfactual GSWP3-W5E5 data for NEU. To obtain the counterfactual annual cycle (orange line), our method aims to map the late factual (thin black line) to the early factual (thick black line) annual cycle.*

**Temperature, Northern Europe (NEU).** There is essentially no difference between the GSWP3 data and the GSWP3-W5E5 data in the period 1979-2014 where the original GSWP3 and W5E5 data overlap. Our approach successfully removes the long-term trend from the observed time series of regionally averaged annual temperature data (Fig. 7(a)) and for each day of the year (Fig. 7(b)). By construction, the detrending retains the year-to-year variability, i.e. hot days stay hot and cold days stay cold. The counterfactual 1990-2019 averages for individual days of the year match the seasonal evolution of the factual data at the beginning of the century (1901-1930) as intended. In Northern Europe, temperatures for each day of the year have changed relatively uniformly throughout the year (Fig. 7(b)).

**Precipitation, Northern Europe (NEU).** The GSWP3 data is offset to slightly higher values of precipitation compared to the GSWP3-W5E5 data in the period 1979-2014 where the original GSWP3 and W5E5 data overlap. The homogenization method of the GSWP3-W5E5 data transfers this offset to the period 1901-1979 leading to a more consistent dataset. Our approach successfully removes the long-term trend from the observed annual regional average time series (Fig. 7(c)). There is a seasonality in the long-term trend with almost no change in April and August in contrast to positive trends in the other months (compare thick to thin black line in Fig. 7(d)). Our approach successfully captures this seasonal variation of the

trend. The annual cycle of the counterfactual data in the period 1990-2019 (orange line) matches the annual cycle of the factual data in the beginning of the century 1901-1930 (thick black line).

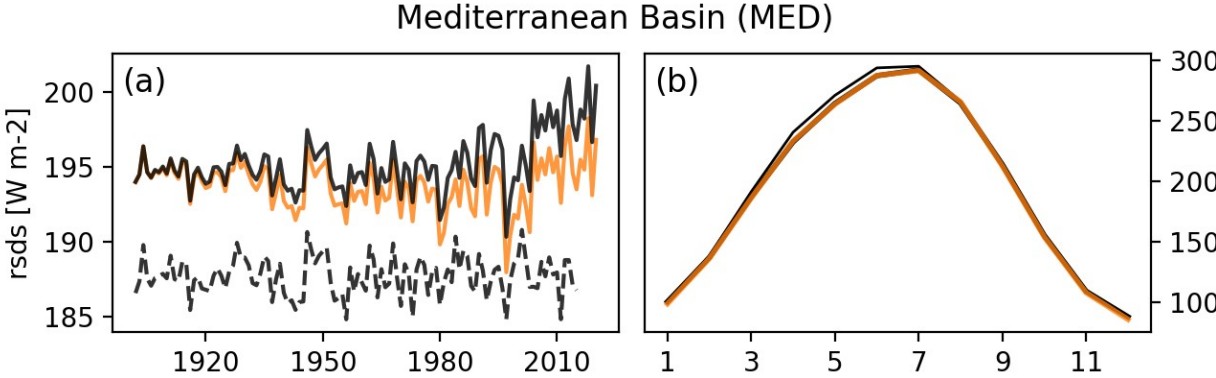

*Figure 8. Same as Fig. 7 but for shortwave radiation over the Mediterranean Basin (MED).*

**Shortwave radiation, Mediterranean Basin (MED).** There is a considerable offset between the GSWP3 and W5E5 data in the overlapping 1979-2014 period (see difference between dashed and solid black line in Fig. 8). In addition, the GSWP3 data do not show a trend over the entire time period 1901-2014 whereas there is a positive trend in the 1979-2019 W5E5

data. The harmonisation has shifted the original GSWP3 data but did not introduce a trend by design of the quantile mapping method used for it (see Sect. 2). This results in inhomogeneous decadal trends in the GSWP3-W5E5 data and a jump at the 1978/1979 transition. This change in the characteristics of the shortwave radiation in GSWP3-W5E5 is an artefact introduced by the different characteristics of the GSPW3 and W5E5 data and not related to GMT change. Thus, in this region the trend in the factual rsds time series is not reliable enough to derive a meaningful 'no climate change'

counterfactual rsds time series. Annual shortwave radiation over Northern Europe is affected in a similar way (Fig. S21).

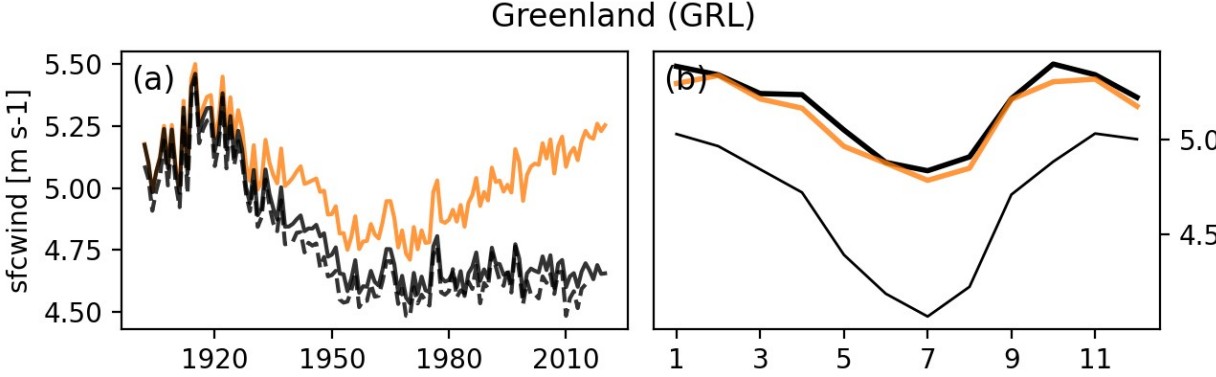

*Figure 9. Same as Fig. 7 but for wind speed over Greenland (GRL).*


**Wind speed, Greenland (GRL).** The factual datasets show spurious trends in wind speed over Alaska, Northern Canada and Greenland over the first half of the 20th century (Figs. 9 and S16), which are related to artificial extratropical cyclone trends in the 20CR reanalysis over that time period (Wang et al. 2013). Shortwave radiation in those regions is affected in a

similar way (Figs. S15 and S17). Our detrending method is unable to distinguish spurious trends from real trends. It finds a correlation between GMT change and the spurious trends and produces counterfactual data that have a spurious positive

trend over the second half of the 20th century (Fig. 9(a)). Such counterfactual time series are clearly not reliable.

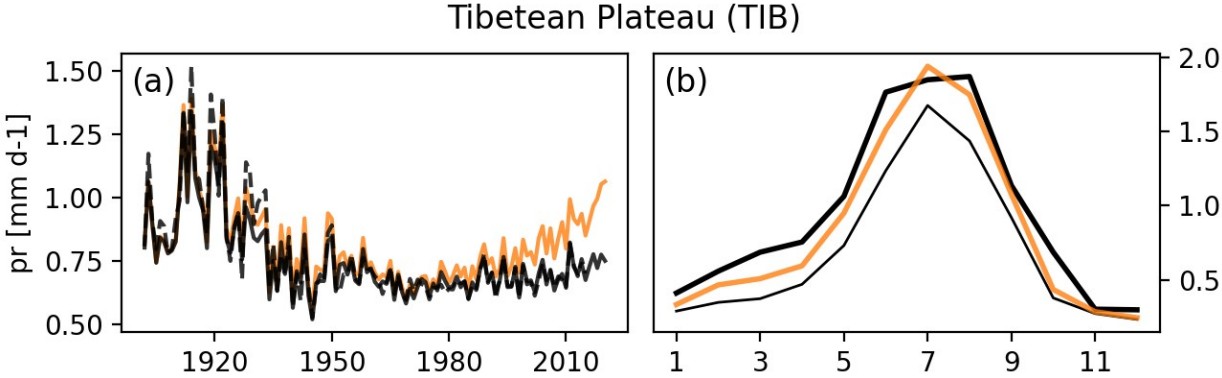

**Figure 10.** Same as Fig. 7 but for precipitation over the Tibetean Plateau (TIB).

**Precipitation, Tibetean Plateau (TIB).** Over the first half of the 20th century the GSWP3-W5E5 precipitation data show a strong drying trend over the TIB region that is assumed to be artificial and inherited from the underlying GPCC dataset (Sect. 2). Since the trend is not related to global warming it is not well captured by our detrending model. Consequently, the average counterfactual precipitation at the end of the observational period does not match the average factual data at the beginning of the period (Figs. 5(h) and 10(b)). The detrending leads to a positive trend over the second half of the century while the factual data do not show such a trend. Since the observational data for the first half of the century are considered unreliable it is also not fit to derive a meaningful 'no climate change' counterfactual.

We present further plots covering all variables and Giorgi regions in the Supplement. Given potential artefacts in the factual data the associated plots have to be analysed when planning a regional attribution study.

**5 Discussion**

The attribution of changes in the climate system to anthropogenic interference with the climate system are a mature research field (IPCC 2013; Gillett et al. 2016; NAS 2016; Stott et al. 2016). Less work has been done on the attribution of changes in natural, human, and managed systems affected by climate change in combination with other time-evolving drivers. Impact attribution as defined in the introduction aims to quantify the role of climate change versus the other drivers of change. Impact attribution needs a comparison of the observed state of the considered system to its hypothetical, counterfactual state without climate change. The reason for the change in climate trends and a separation of anthropogenically forced changes from climate variability are not necessarily required. Thus, a simplified methodology that detrends observational data is sufficient without the need for probabilistic climate model simulations. The proposed design of the counterfactual climate forcing data and the associated impact simulation framework mean a restriction to 'impact attribution to climate change' instead of 'impact attribution to anthropogenic climate forcing'. The latter is necessary to, for example, attribute a fraction of an impact to a greenhouse gas emitter and support climate litigation (Marjanac, Patton, and Thornton 2017; Burger, Wentz, and Horton 2020). Thus the counterfactual climate data generated here are not intended to replace climate simulations with counterfactual greenhouse gas forcings such as the histNAT CMIP6 experiments (Gillett et al. 2016) large climate model ensembles that are required to attribute changes in climate or impacts to anthropogenic emissions.

Climate impact models can be considered as ideal tools to address impact attribution as they are usually designed to represent the response of impact indicators to climate disturbances but also account for direct human interventions such as agricultural management changes, water abstraction or flood protection measures. Within the model, individual drivers can be controlled and a factual run (observed climate change + observed direct human interventions, Fig. 2 blue line) can be compared to a counterfactual run (counterfactual climate + observed direct human interventions, Fig. 2 orange line).

By providing climate forcing data for counterfactual climate impact runs, we facilitate impact attribution following the basic IPCC AR5 WGII definition utilizing the strength of impact models to address the important question to what degree climate change is already affecting natural, human and managed systems. So far the contribution of climate change to long-term historical changes in human, natural or managed systems is often addressed by model simulation where direct human interventions are fixed while only climate is allowed to change according to historical observations (e.g. Sauer et al. 2021). However, this alternative definition may also lead to different results and does not allow for the attribution of the magnitude of individual impact events to climate change as described in Fig. 2.

Attribution draws a causal connection and quantifies the change due to the cause. An important part of the attribution work is thus to ensure that the cause-effect relationship is correctly captured in the model. This requires careful analysis and model evaluation to show that the change estimated by an impact model is a reliable estimate of the real-world change. Simulated changes need to agree with observed changes and it needs to be ruled out that this agreement is due to confounding factors that drive observed change, but are not part of the model simulations. The ISIMIP3a historical simulations serve to address these points and demonstrate the explanatory power of impact models as an integral part of the attribution work.

Our method ultimately builds on the correlation between a regional climate variable and decadal GMT change to remove long-term trends in the regional climate variables without implying causality. It is well possible that changes in regional climate variables have other reasons than global warming such as local effects of land use changes and aerosol emissions as well as regional characteristics of large scale decadal climate oscillations. However, our study shows that GMT change is generally a powerful predictor allowing for generating stationary counterfactual climate data. Major detrending failures seem to be related to artefacts in the factual observational climate data that particularly affect the first half of the century and prevent impact attribution in the proposed framework.

Our detrending approach does not guarantee the maintenance of physical consistency of different climate variables in the counterfactual datasets in terms of, e.g., energy closure or water budgets. However, the applied quantile mapping preserves ranks, which means that relatively high values before the mapping are also relatively high after the mapping and similarly for relatively low values. Statistically speaking, univariate quantile mapping independently applied to all climate variables preserves the multivariate rank distribution (the copula) over all variables. In that sense the statistical dependence between variables is preserved by our detrending method and the risk of producing physically inconsistent counterfactual climate data is at least limited. This is critical for the attribution of the extreme event magnitude to observed climate trends (see introduction) because several climate variables can contribute to impact extremes.

Here, we deliberately excluded the question of what drives climate change, i.e. the attribution of changes in the climate system to greenhouse gas emissions, as it often implies a focus on this aspect and less attention is paid to the separation of climate change from direct human interventions as drivers of observed changes in natural, human and managed systems. The restriction of the research question to 'impact attribution to climate change in general' makes the assessments independent of

climate simulations and their potential limitation in reproducing processes relevant for historical climate change. Instead, the restricted framework is directly linked to impact model evaluation and the question of how well we understand the observed changes in human, natural, and managed systems. This question can most directly be addressed by the factual impact simulations proposed here rather than with impact simulations based on simulated historical climate. In addition, as opposed to large ensembles of climate model simulations, such a dataset is easily integrated into an impact model intercomparison project such as ISIMIP, which includes models of very different computational costs. In this way the approach allows for an exploration of structural uncertainty in climate impact attribution, based on a multi-impact-model ensemble, combined with a variety of damage functions where appropriate.

With the methods and data presented here, we aim to advance the field of impact attribution and reveal past and present societal and environmental sensitivities to climate change. Getting a better understanding of the drivers of observed changes in natural, human and managed systems will help us to better estimate future risks related to ongoing global warming and develop adequate adaptation measures.

**Code and data availability.**
The source code underlying the analysis presented in the paper (v1.1.0) is archived at
https://doi.org/10.5281/zenodo.5032065. The source code to produce the figures as appearing in the manuscript (v1.1.0) is archived at https://doi.org/10.5281/zenodo.5036701. All code is open to use under the GPL license. The presented counterfactual climate dataset is archived at https://doi.org/10.5281/zenodo.5036364 and based on v1.1.0 of the source code.

**Author contributions.**
MM, ST, SL and KF developed the concept. ST and MM implemented the methods, wrote the code and produced the data. All authors wrote the paper.

**Competing interests.**
The authors declare that they have no conflict of interest.

**Acknowledgements.** This work was funded by the German Federal Ministry of Education and Research (BMBF, grant agreement 01LS1711A). Factual and counterfactual GSWP3-W5E5 data are made available through the ISIMIP project (www.isimip.org). S. L. acknowledges funding from the European Union's Horizon 2020 research and innovation programme under grant agreement No. 641816 (CRESCENDO). S. T. acknowledges funding from the European Union's Horizon 2020 research and innovation programme under grant agreement No. 820712 (RECEIPT). We thank Hyungjun Kim for helping us to explain the making of the GSWP3 dataset. We thank Anne Gädeke and Christoph Menz for beta-testing our counterfactual data. We are grateful to Benjamin Schmidt early contributions to the code. We thank the two anonymous reviewers for their helpful comments on the initial version of this manuscript.

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
