# Peer review of "ATTRICI v1.1 - counterfactual climate for impact attribution"

_Geoscientific Model Development, 2020_

## Referee Comment (RC1) · Anonymous Referee #1 · 26 Aug 2020

This paper describes the construction of so-called counterfactual datasets, using parametric statistical models to describe high frequency variability in the observed climate, removing trend from the parameters of those distributions via what is essentially a low-pass filtering approach, and then mapping observed quantiles onto the quantiles of the distributions with trend removed. Some basic results are show illustrating the removal of trend and the retention of variability. The method is applied to two gridded datasets.

Things that I found to be missing from this paper include 1) Justification for the various choices that are made in setting up the probability model (e.g., why Gaussian for several variables, Weibull for wind speed, and why the particular choice of low-pass filtering approach based on singular spectrum analysis). 2) Motivation for the decision to fit the model using a Bayesian approach, which then requires that priors of various

kinds be chosen – and justified. 3) Discussion about how one goes about ensuring that dependence between variables is respected (which needs to be well represented for many impacts models). 4) Discussion about maintaining physical consistency between variables (also required by many impacts models, e.g., to ensure closure of energy and water budgets) and how it is maintained in the heavily processed counterfactual datasets that are produced. 5) Sufficient discussion of the homogeneity (or lack thereof) of the underlying factual dataset and its impacts. The authors attempt to minimize potential problems, for example, by pointing to a basis in a long reanalysis (line 70) and by trying to argue that early data have limited influence (lines 282-283), but I don't find these arguments very convincing. 6) Cross-validation using references from the literature concerning differences between factual and constructed counterfactual climates that are found. For example, have others written about changes in South American wind speeds, and have causes been explored and tested in models? Surely, this kind of validation is the minimum that should be expected to ensure that the datasets that are produced are fit-for-use. 7) Discussion of what applications might, or might not be suitable.

There are also some aspects that I found to be somewhat confusing. An example is the consideration of daily temperature skewness – which I found confusing given that daily mean temperature is modelled as a Gaussian random variable (and thus has zero skewness, by assumption). What is being discussed using some kind of shorthand that is known to the authors, but perhaps not others, is the nature of the diurnal temperature cycle – but how that diurnal cycle, and its variation in time, is represented by daily minimum temperature, daily maximum temperature and a measure of "skewness" of some kind is not made clear. Another example is the choice to represent relative humidity (which has a strong diurnal cycle that can be important for some impacts models, and is confined to values between 0 and 100%) as a Gaussian variable – and then to clip that distribution if quantile mapping happens to produce values outside the 0-100% interval.

Finally, it seems to me that this paper is not a terribly good fit for GMD; it might be a better suited for a data journal in my view.

---

## Referee Comment (RC2) · Anonymous Referee #2 · 27 Aug 2020

The authors present a new methodology to derive a counterfactual climate, relevant for impact studies. This is a very relevant and topical area to be looking at, and the PIK team are world leading in this area, so I was excited to see this paper from their team. However, I do think the paper is not exactly up to their usual standard, and in particular a thorough understanding of the attribution science is not clear in this paper.

My main concern is the framing and interpretation of the question being posed. I have to confess that it took me reading the entire paper to fully understand what they were doing, and how it is different to other methods. I think the authors need to work on posing the problem much earlier, and discussing how it fits in to the wider attribution methods. In IPCC WG1 attribution is often split into trend attribution, and event attribution – the authors mention this at one point, but do not really explain how their

methodology fits into this concept. I was also very surprised not to see any of the available attribution impact work cited in this paper. I feel this is a very large omission from the paper, and the authors need a paragraph or two, maybe in their discussion section, introduction section, or both, that describes these papers, and explain how the author's current views and concepts fit into that. From the top of my head, these papers are all very relevant: Hydrology: Pall et al, 2011, Schaller et al, 2014; Health: Astrom et al, 2013, Mitchell et al, 2016; economy: Frame et al, 2020.

In a similar context, I also believe the authors need to highlight the weaknesses in their approach, as compared to other approaches. Two obvious ones are: 1) any attempt at making a counterfactual climate is difficult, and can be done multiple ways. Many other methods therefore provide an uncertainty in their modelled climate, but you do not. I understand why you don't, but the implications of this are important. 2) Many authors have argued that impacts are felt in the extremes of climate more so than elsewhere, that is why counterfactual attempts are often made with very large model simulation sizes. This is not so easy in your methodology, although I can see ways forward for it – this should be discussed.

The authors have worked in IPCC WG2 for a long time, and maybe a bit in WG1, but they need to be aware that their readers might be solely in one WG (or none at all), so the subject specific language needs to be very simple for a paper like this.

Other corrections

• Title "counterfactual climate for impact attribution" – I see why you have this title, but it seems that your work would be very useful • Line 9: "anthropogenic" is needed before climate change • Line 9: "Other drivers change according to observations". Actually I think the other drivers should remain the same according to observations. • Line 19-21: This sentence is very confusing without reading the paper, I suggest making it stand alone. • Line 26: Citation needed. Haustien et al, 2017 is a good one, but there are others. • Figure 1: In this figure you show "climate change"

[Figure]

as the affected quantity – this should just be "climate". Likewise for the driver in the second panel. I also do not agree with the caption that this is how the IPCC frame attribution. • Lines 31-38: I like this description, and it is now clearer in my head what you are doing. If this section can be summed up on 1 line for the abstract, that would really help make things clear from the start. • Line 63-66: This section is confusing me, in much the same way the end of the abstract did. Specifically, you say impact attribution does not need to address the causes of climate change. So, what is it addressing? You could state that explicitly here. I also think the attribution community would see this differently, and there is a danger that people will now be confused over what this term means. • Data section: What is the spatial scale of the dynamically downscaled data? How much do we trust this data, especially in poorly observed parts of the world? What are the implications for these problems on the questions posed? • Line 101: Lots of work has been done on pattern scaling recently, so I think a more modern approach should be cited here. E.g. Herger, 2015, although many others exist as well. • Line 101-105: This paragraph makes it much clearer in my head what you are doing. I would use some of this text to explain this earlier, especially about the non-causal link with global temperature. • Page 7: Why are the distribution names bolded? • Line 144: I understand why you are including hurs, and commend it, but it is a bounded quantity (i.e. nominally constrained between 0-100), so would that cause problems when modelling with a Gaussian? • Figures 4 and 5: These are very nice and informative figures. • Lines 255-256: I agree it is rare, but there are still numerous studies that have done this (see major comment). • Line 277-279: You should state clearly here why your data is useful in a complementary way to Gillett et al

---

## Author Comment (AC1) · 11 Jan 2021

Dear referee,

many thanks for your helpful comments. Please find our responses in the attached supplementary PDF. The document comprises responses to both referee 1 and referee 2. Responses to referee 2 start at page 9.

On behalf of the author team, yours sincerely Matthias Mengel

Please also note the supplement to this comment:
https://gmd.copernicus.org/preprints/gmd-2020-145/gmd-2020-145-AC1-supplement.pdf

---

## Author Comment (AC2) · 11 Jan 2021

*Referee comments in italic*
Responses in blue
"Quotes from the revised manuscript as plain text"

**Anonymous Referee #1**

##

*This paper describes the construction of so-called counterfactual datasets, using parametric statistical models to describe high frequency variability in the observed climate, removing trend from the parameters of those distributions via what is essentially a lowpass filtering approach, and then mapping observed quantiles onto the quantiles of the distributions with trend removed. Some basic results are show illustrating the removal of trend and the retention of variability. The method is applied to two gridded datasets.*

We kindly thank the referee for the evaluation of the manuscript. Due to the substantial comments from both referees, we substantially overhauled both manuscript and methodology.

*Things that I found to be missing from this paper include*
*1) Justification for the various choices that are made in setting up the probability model (e.g., why Gaussian for several variables, Weibull for wind speed, and why the particular choice of low-pass filtering approach based on singular spectrum analysis).*

The manuscript now includes a justification for the choice of each distribution in section 3.2, see lines 330 to 364. We based most of our choices on existing work from the field of bias correction. For example, using Weibull to model the wind speed distribution is a common choice for parametric quantile mapping of that variable (e.g., (Lange 2019; Li et al. 2019)). We use singular spectrum analysis (SSA) to smooth (or low-pass filter) the predictor time series of global mean temperature because it is a widely applied and long-standing method in the field, see for example (Ghil and Vautard 1991), (Schlesinger and Ramankutty 1994) or (Rahmstorf 2007) for applications and (Ghil et al. 2002) for a review of the method. We have also added the original GMT time series to Figure 3 (former Figure 2) to show the effect of the smoothing. We only smooth the predictor and not the spatial climate data from which the counterfactual is derived. We now write, see lines 244 to 258:

"Here, $T$ is GMT change since 1901 smoothed by singular spectrum analysis (Michael Ghil et al. 2002) with a smoothing window of 10 years (Fig. 3). The smoothing of the predictor is applied because we only want to remove long-term trends from the regional climate time series. Natural climate variability on shorter time scales due to phenomena such as the El Niño–Southern Oscillation is retained. Using $T$ as the predictor means that we remove long-term trends in regional climate to the extent that those are correlated with GMT change,

but irrespective of the cause of global warming."

*2) Motivation for the decision to fit the model using a Bayesian approach, which then requires that priors of various kinds be chosen -- and justified.*

We now provide a motivation and justify our prior choices, see lines 297 to 306:

"We use a Bayesian approach to estimate all of these parameters. This requires the specification of prior distributions of the model parameters. Similar to regularization techniques in frequentist approaches, the prior allows to focus the model fitting on plausible parameter values. This is particularly important for numeric stability when the logit and logarithm link functions are applied. We use a zero-centered Gaussian prior for all parameters and all climate variables because we normalize the data before parameter estimation. We use a standard deviation of 1.0 for $a_0^{(intercept)}$, a standard deviation of $1/(2k-1)$ for $a_k^{(intercept)}$, $k = 1, ..., 4$, and a standard deviation of 0.1 for $a_k^{(slope)}$, $k = 0, ..., 4$. Our choice of priors for $a_k^{(intercept)}$ is based on the assumption that the first mode with a period of one year explains the largest part of the annual cycle and higher-order modes have decreasing influence. However this is only a prior assumption, i.e. if the data show different patterns, they can still be captured by our model. For $a_k^{(constant)}$ we use the same priors as for $a_k^{(intercept)}$. We use the same priors for the parameters $b_k$."

We reworked our model formulation, now fitting into the class of Generalized Linear Models, to unify it across variables and make it easier to grasp for the reader. This also influenced choices of priors.

*3) Discussion about how one goes about ensuring that dependence between variables is respected (which needs to be well represented for many impacts models).*

We use quantile mapping for detrending the time series. Quantile mapping preserves ranks, which means that relatively high values before the mapping are also relatively high after the mapping and similarly for relatively low values. This is now explained in the main text, see lines 507 to 514:

"Our detrending approach does not guarantee the maintenance of physical consistency of different climate variables in the counterfactual datasets in terms of, e.g., energy closure or water budgets. However, the applied quantile mapping preserves ranks, which means that relatively high values before the mapping are also relatively high after the mapping and similarly for relatively low values. Statistically speaking, univariate quantile mapping independently applied to all climate variables preserves the multivariate rank distribution (the copula) over all variables. In that sense the statistical dependence between variables is preserved by our detrending method and the risk of producing physically inconsistent counterfactual climate data is at least limited. This is critical for the attribution of the extreme event magnitude to observed

climate trends (see introduction) because several climate variables can contribute to impact extremes.”

*4) Discussion about maintaining physical consistency between variables (also required by many impacts models, e.g., to ensure closure of energy and water budgets) and how it is maintained in the heavily processed counterfactual datasets that are produced.*

That is an important point. While our approach preserves statistical dependence between variables (see previous point) it does not imply maintenance of physical consistency. This is similar to the inconsistencies introduced by bias correction methods, which are widely applied in the field. Yet we think that the preservation of statistical dependence at least limits the risk of producing physically inconsistent counterfactual climate data (see argument given above).

*5) Sufficient discussion of the homogeneity (or lack thereof) of the underlying factual dataset and its impacts. The authors attempt to minimize potential problems, for example, by pointing to a basis in a long reanalysis (line 70) and by trying to argue that early data have limited influence (lines 282-283), but I don't find these arguments very convincing.*

The factual climate data are certainly subject to regional artefacts that also affect the counterfactual data. In the revised manuscript we now make very explicit that the counterfactual data should not be used blindly for the attribution of historical changes in natural, human or managed systems. Instead we recommend a number of control plots to check for potential artefacts in the observed trends in climate variables. For brevity, we now present counterfactuals only for the GSWP3-W5E5 dataset. We now write, see lines 370 to 375:

“In addition, we propose to plot the entire time series for regionally averaged annual (or seasonal) mean values for both the original and the counterfactual climate data. Here, we do so for annual regional averages over 21 world regions (Giorgi and Francisco 2000), see left panels of Figs. 7-10 and supplementary figures, but propose to adjust the regions and season for each attribution study individually according to its focus. For our specific observational dataset we add annual regional averages of the original GSWP3 data to check if the homogenisation of GSWP3 with W5E5 has introduced artificial trends in the factual GSWP3-W5E5 data.”

In the paper we provide these plots for all world regions defined by (Giorgi and Francisco 2000) and all climate variables covered. The results section now includes an exemplary discussion for a subset of regions and variables. In addition, we highlight already in the introduction that each impact attribution study based on the proposed factual and counterfactual climate forcing first needs an evaluation of the factual impact simulation that additionally reduces the risk of erroneous attribution of impacts, see lines 105 to 112 and the newly added Fig. 2:

“In a first step the climate impact model forced by observed climate and socio-economic drivers has to demonstrate to be able to reproduce the observed changes in natural, human and managed systems as measured by an impact indicator (comparison of black and blue solid lines

in Figure 2). The attribution of the observed changes in natural, human and managed systems is built on a high explanatory power of the factual simulations. Then, in a second step, that factual simulation can be compared to a counterfactual simulation, forced by counterfactual climate but otherwise the same input as in the factual simulation. Such a comparison allows for a quantification of the contribution of climate change to both the observed trend in the impact indicator (CT in Fig. 2) and the observed magnitude of an individual impact event (CE in Fig. 2)."

and lines 140 to 142:

"Low-quality factual climate forcing data is expected to result in a low-quality reproduction of observed variations in the impact indicators of interest. If that is the case, the simulation set-up outlined here does not allow for an attribution of the observed changes in impacts to climate change."

*6) Cross-validation using references from the literature concerning differences between factual and constructed counterfactual climates that are found. For example, have others written about changes in South American wind speeds, and have causes been explored and tested in models? Surely, this kind of validation is the minimum that should be expected to ensure that the datasets that are produced are fit-for-use.*

The focus of our paper is on the detrending approach and less on the detailed evaluation of the factual climate forcing data. This is now more explicitly stated in the text:

"In this paper we introduce a detrending method tailored to support impact attribution and illustrate its application to one of the observational climate datasets provided within ISIMIP3a (https://protocol.isimip.org/protocol/ISIMIP3a, see data section below). The quality of the associated impact attribution studies will critically depend on the quality of that observational dataset. Deficits in the observational data may lead to artefacts in the derived historical trends. For the dataset used here, we identify some of those artifacts. Since it is expected that other artifacts will be found for other observational datasets, impact attribution studies should ideally be based on a range of different observational datasets to facilitate a quantification of the contribution of observational climate data uncertainty to the uncertainty of the attribution results. This is also planned within ISIMIP3a."

Other observational datasets that could also be used in the described attribution set-up are the Princeton data (PGMFD v2.1, Sheffield, Goteti, and Wood 2006) and the WATCH data (WFD, Weedon et al. 2011), and the combined data set WATCH-WFDEI (Weedon et al. 2011, 2014) that have already been used within ISIMIP2a. We make our methods freely available, so they can readily be applied to other datasets to explore data uncertainties in impact attribution results.

To make it easier to judge the fitness of the GSWP3-W5E5 data for attribution, we expanded

the Data section and mention the deficits of the dataset more explicitly, see e.g. lines 191 to 198:

"Examples of temporal inhomogeneities in GSWP3 that are relevant for this study include artificial drying trends over northwest China and the Tibetan Plateau over the first half of the 20th century (Fig. 10) that are inherited from GPCC (Chen and Frauenfeld 2014), and spurious trends in shortwave radiation and wind speed over Alaska, Northern Canada and Greenland over the first half of the 20th century (Figs. 9 and S12), which are related to artificial extratropical cyclone trends in 20CR over that time period (Wang et al. 2013). Generally, the quality of 20CR, and hence GSWP3, becomes relatively stable around mid-century over the Northern Hemisphere, earlier over Europe and later over the Southern Hemisphere, in line with variations in the availability of pressure observations for data assimilation in the reanalysis (Compo et al., 2011)."

We additionally exemplify major deficits in the result section.

It is a critical part of attribution to evaluate the impact model simulation against the observed changes in natural, human or managed systems (see response to previous comment) and identify the best data set for specific regions or variables. We encourage evaluation tailored to the specific region and variable, see the new section 3.3, lines 366 to 378:

"To evaluate the detrending method and the counterfactual GSWP3-W5E5 data we use the difference between multi-year averages of each climate variable over the beginning of the time period (1901-1930) and multi-year averages over the end of the time period (1987-2016) as a measure of the trend. We compare this trend measure between the observed data and the counterfactual data, for which it should be close to zero (Figs. 5 and 6). In addition, we propose to plot the entire time series for regionally averaged annual (or seasonal) mean values for both the original and the counterfactual climate data. Here, we do so for annual regional averages over 21 world regions (Giorgi and Francisco 2000), see left panels of Figs. 7-10 and supplementary figures, but propose to adjust the regions and season for each attribution study individually according to its focus. For our specific observational dataset we add annual regional averages of the original GSWP3 data to check if the homogenisation of GSWP3 with W5E5 has introduced artificial trends in the factual GSWP3-W5E5 data. To evaluate the performance of the detrending method for each day of the year we propose to compare the 1987-2016 regional mean climatology of the counterfactual data to the 1901-1930 regional mean climatology of the factual data for each region of interest (right panels of Figs. 7-10 and supplementary figures)."

*7) Discussion of what applications might, or might not be suitable.*

In principle the application of the approach is only limited by our ability to reproduce, i.e. to explain the observed changes in the considered human natural and managed systems. Explanatory power could be constrained by the quality of the observational climate data,

insufficient knowledge about direct (human) forcings, deficits in the process understanding implemented in the impact models or deficits of the observational impact data.

Concretely, our main aim is to foster impact attribution through the ISIMIP project. ISIMIP builds on a protocol of experiments with curated forcing data and covers a range of sectors (global and regional water models, global crop models, global biomes and regional forestry models, models describing energy demand and supply, biodiversity models, permafrost, health, lake and fire models).

Our approach is restricted to impact attribution to climate change in contrast to impact attribution to anthropogenic climate forcing. This is now additionally highlighted in the main text, see lines 61 to 73:

"In addition to 'climate attribution', research on 'impact attribution' addresses the question: To what degree are observed changes in natural, human and managed systems induced by observed changes in climate (Fig. 1, second arrow). In the WGII contribution to the IPCC-AR5, an entire chapter was dedicated to the topic including the following definition: An impact of climate change is 'detected' if the observed state of the system differs from a counterfactual baseline that characterizes the system's behavior in the absence of changes in climate (IPCC 2014, chap. 18.2.1) and 'attribution' is the quantification of the contribution of climate change to the observed change in the natural, human or managed system. In both cases "climate change refers to any long-term trend in climate, irrespective of its cause" (IPCC 2014, chap. 18.2.1).

While in principle changes in natural, human and managed systems could also be attributed to anthropogenic climate forcing ('impact attribution to anthropogenic climate forcing', first and second arrow in Fig. 1, (Pall et al. 2011; Schaller et al. 2016; Mitchell et al. 2016)), we focus on 'impact attribution to climate change' as described in the WGII definition and introduce a climate dataset that can be used as input to climate impact models to characterize the system's behavior in the absence of climate change."

*There are also some aspects that I found to be somewhat confusing. An example is the consideration of daily temperature skewness – which I found confusing given that daily mean temperature is modelled as a Gaussian random variable (and thus has zero skewness, by assumption). What is being discussed using some kind of shorthand that is known to the authors, but perhaps not others, is the nature of the diurnal temperature cycle – but how that diurnal cycle, and its variation in time, is represented by daily minimum temperature, daily maximum temperature and a measure of "skewness" of some kind is not made clear.*

Our time resolution is daily, i.e. the diurnal temperature cycle is not present in the temperature timeseries itself. The distribution of the daily mean temperatures (tas) is in fact not considered skewed. However, the *sub-daily* variation of temperatures can be derived from the daily maximum temperatures (tasmax) and daily minimum temperatures (tasmin) and allowed to be

skewed in the sense that tasskew = (tas - tasmin) / (tasmax - tasmin) may deviate from 0.5. We have now clarified this part in the text, see lines 334 to 341:

"**Daily minimum and maximum near-surface air temperature.** They provide a measure of the diurnal temperature cycle in the daily resolved dataset. We do not estimate counterfactual time series for tasmin and tasmax directly to avoid large relative errors in the daily temperature range as pointed out by (Piani et al. 2010). Instead we construct counterfactuals for the auxiliary variables tasrange = tasmax - tasmin and tasskew = (tas - tasmin) / tasrange that then determine the tasmin and tasmax counterfactuals (Piani et al. 2010). We use the Gamma distribution to model tasrange since it has a lower bound at zero. The expected value is modeled according to Eq. (1). The skewness of daily near-surface temperature tasskew is modeled by a Gaussian distribution. While theoretically bounded, tasskew is never close to its bounds of zero and one. This justifies the Gaussian model choice."

*Another example is the choice to represent relative humidity (which has a strong diurnal cycle that can be important for some impacts models, and is confined to values between 0 and 100%) as a Gaussian variable – and then to clip that distribution if quantile mapping happens to produce values outside the 0-100% interval.*

Based on the comment we reworked the model and now use a beta distribution to describe the relative humidity. We now write, see lines 360 to 363:

"Near-surface relative humidity *hurs* is positive and less than or equal to one. We assume *hurs* to follow a beta distribution. Its expected value is allowed to vary with T and t. The associated coefficients are estimated using a beta regression model (Ferrari and Cribari-Neto 2004) and Eq. (1) for the expected value while the dispersion parameter, $\varphi$ , is assumed to only vary with t."

The diurnal cycle of relative humidity is not covered in our input datasets. Only the daily mean relative humidity is included and we therefore only provide counterfactuals for the daily mean.

*Finally, it seems to me that this paper is not a terribly good fit for GMD; it might be a better suited for a data journal in my view.*

The focus of the paper is actually on the detrending method itself and on the introduction of the general framework for impact attribution within the third phase of the Intersectoral-Impact model Intercomparison Project ISIMIP3a. In the revised version of the manuscript we only discuss one of the datasets that were included in the original submission (GSWP3-W5E5) as an example application. The illustrative character has been made more explicit in the main text, see lines 128 to 135:

"In this paper we introduce a detrending method tailored to support impact attribution and illustrate its application to one of the observational climate datasets provided within ISIMIP3a (https://protocol.isimip.org/protocol/ISIMIP3a, see data section below). The quality of the associated impact attribution studies will critically depend on the quality of that observational dataset. Deficits in the observational data may lead to artefacts in the derived historical trends. For the dataset used here, we identify some of those artifacts. Since it is expected that other artifacts will be found for other observational datasets, impact attribution studies should ideally be based on a range of different observational datasets to facilitate a quantification of the contribution of observational climate data uncertainty to the uncertainty of the attribution results. This is also planned within ISIMIP3a."

We refer to the editor to make the decision of journal fit.

**Anonymous Referee #2**

*The authors present a new methodology to derive a counterfactual climate, relevant for impact studies. This is a very relevant and topical area to be looking at, and the PIK team are world leading in this area, so I was excited to see this paper from their team. However, I do think the paper is not exactly up to their usual standard, and in particular a thorough understanding of the attribution science is not clear in this paper.*

We thank the reviewer for this thorough and helpful review. We modified the text to better embed our approach into the context of the existing literature of attribution science, in particular how our work relates to the definition of attribution as used in IPCC WGI and WGII.

*My main concern is the framing and interpretation of the question being posed. I have to confess that it took me reading the entire paper to fully understand what they were doing, and how it is different to other methods. I think the authors need to work on posing the problem much earlier, and discussing how it fits into the wider attribution methods. In IPCC WG1 attribution is often split into trend attribution, and event attribution – the authors mention this at one point, but do not really explain how their methodology fits into this concept.*

This is absolutely right, and we thank the reviewer for this important remark. We now outline our framing of attribution in the abstract and we have reworked the introduction including Figure 1 and added a new Figure 2. We clarify that the approach is designed to support 'impact attribution to climate change' in the WGII sense and point out that this perspective differs from the WGI lens. The main question being posed is: *how large is the influence of climatic trends on observed changes in natural, human and managed systems that also respond to other time-evolving drivers of change?* We aim to allow for the quantification of the contribution of climate trends in comparison to potential other non-climate related drivers. The influence of anthropogenic forcing to the climate trend is not investigated. We now write, see lines 61 to 76:

"In addition to 'climate attribution', research on 'impact attribution' addresses the question: To what degree are observed changes in natural, human and managed systems induced by observed changes in climate (Fig. 1, second arrow). In the WGII contribution to the IPCC-AR5, an entire chapter was dedicated to the topic including the following definition: An impact of climate change is 'detected' if the observed state of the system differs from a counterfactual baseline that characterizes the system's behavior in the absence of changes in climate (IPCC 2014, chap. 18.2.1) and 'attribution' is the quantification of the contribution of climate change to the observed change in the natural, human or managed system. In both cases "climate change refers to any long-term trend in climate, irrespective of its cause" (IPCC 2014, chap. 18.2.1).

While in principle changes in natural, human and managed systems could also be attributed to anthropogenic climate forcing ('impact attribution to anthropogenic climate forcing', first and second arrow in Fig. 1, (Pall et al. 2011; Schaller et al. 2016; Mitchell et al. 2016)), we focus on 'impact attribution to climate change' as described in the WGII definition and introduce a climate dataset that can be used as input to climate impact models to characterize the system's behavior in the absence of climate change. As the response of natural, human or managed systems to climate and socio-economic forcings is commonly considered deterministic (or at least simulated in this way) the attribution of impacts to the observed realisation of climate change does not necessarily have to be addressed in a probabilistic way and gives more weight to the separation of climate change from direct human influences as potential drivers of changes in the considered systems."

*I was also very surprised not to see any of the available attribution impact work cited in this paper. I feel this is a very large omission from the paper, and the authors need a paragraph or two, maybe in their discussion section, introduction section, or both, that describes these papers, and explain how the author's current views and concepts fit into that. From the top of my head, these papers are all very relevant: Hydrology: Pall et al, 2011, Schaller et al, 2014; Health: Astrom et al, 2013, Mitchell et al, 2016; economy: Frame et al, 2020.*

We revised the manuscript to better embed the proposed approach into existing work on impact attribution, see lines 62 to 73:

"In the WGII contribution to the IPCC-AR5, an entire chapter was dedicated to the topic including the following definition: An impact of climate change is 'detected' if the observed state of the system differs from a counterfactual baseline that characterizes the system's behavior in the absence of changes in climate (IPCC 2014, chap. 18.2.1) and 'attribution' is the quantification of the contribution of climate change to the observed change in the natural, human or managed system. In both cases "climate change refers to any long-term trend in climate, irrespective of its cause" (IPCC 2014, chap. 18.2.1).

While in principle changes in natural, human and managed systems could also be attributed to anthropogenic climate forcing ('impact attribution to anthropogenic climate forcing', first and second arrow in Fig. 1, (Pall et al. 2011; Schaller et al. 2016; Mitchell et al. 2016)), we focus on 'impact attribution to climate change' as described in the WGII definition and introduce a climate dataset that can be used as input to climate impact models to characterize the system's behavior in the absence of climate change. "

as well as lines 114 to 127:

"Process-based impact models such as those taking part in the ISIMIP project (www.isimip.org) are ideal tools to address impact attribution as they generally describe the response of natural,

human or managed systems not only to climate but also direct (human) drivers. For example, crop models can simulate the response of crop yields to changes in land use, irrigation patterns, fertilizer input and crop varieties (Lobell, Schlenker, and Costa-Roberts 2011; Challinor et al. 2014; Minoli et al. 2019). Similarly, hydrological models can be used to simulate how dam construction and water withdrawal affect river discharge (Veldkamp et al. 2017, 2018). In addition, those models allow for a process-based representation of the extent of e.g. river floods and droughts that can be combined with maps of asset distribution and empirical damage functions to estimate the direct economic damages induced by weather extremes. The impact attribution framework could then be used to approximate the contribution of climate change to observed trends in reported damages. Using process-based climate impact models, this contribution can be explicitly separated from changes in damages driven by changes in exposure or vulnerability. In this regard it goes beyond available approaches of damage attribution that attribute to anthropogenic climate forcing but simply estimate the contribution of anthropogenic climate forcing to observed damages by multiplying the fraction of attributable risk associated with weather extremes by the observed damage (Frame et al., 2020). In the same way, it could improve the attribution of health impacts (Mitchell et al. 2016).”

We do not refer to Astrom et al. (2013) as their work is on future health impacts and not on observed historical changes.

*In a similar context, I also believe the authors need to highlight the weaknesses in their approach, as compared to other approaches. Two obvious ones are:*

1)  *any attempt at making a counterfactual climate is difficult, and can be done multiple , ways. Many other methods therefore provide an uncertainty in their modelled climate, but you do not. I understand why you don't, but the implications of this are important.*

We consider the potential deficits of the observational climate data as the main source of uncertainty in the proposed impact attribution framework and propose to address this aspect by using more than one factual-counterfactual pair of climate forcing data for impact attribution, see lines 129 to 142:

“In this paper we introduce a detrending method tailored to support impact attribution and illustrate its application to one of the observational climate datasets provided within ISIMIP3a (https://protocol.isimip.org/protocol/ISIMIP3a, see data section below). The quality of the associated impact attribution studies will critically depend on the quality of that observational dataset. Deficits in the observational data may lead to artefacts in the derived historical trends. For the dataset used here, we identify some of those artifacts. Since it is expected that other artifacts will be found for other observational datasets, impact attribution studies should ideally be based on a range of different observational datasets to facilitate a quantification of the contribution of observational climate data uncertainty to the uncertainty of the attribution results. This is also planned within ISIMIP3a. For the dataset considered here and potential additional

ones we propose a collection of control plots that should be used to scan the observational climate data for artifacts in preparation of each individual impact attribution study. While we provide the control plots for a set of large-scale world regions and all climate variables covered by our observational climate dataset, they should be adjusted to the regions and variables of interest in an impact attribution study as part of the analysis of the factual impact simulation (Fig. 2). Low-quality factual climate forcing data is expected to result in a low-quality reproduction of observed variations in the impact indicators of interest. If that is the case, the simulation set-up outlined here does not allow for an attribution of the observed changes in impacts to climate change."

We realized that it is important to better understand our attribution set-up and the role of climate therein. We now better describe the set-up in lines 61 to 67:

In addition to 'climate attribution', research on 'impact attribution' addresses the question: To what degree are observed changes in natural, human and managed systems induced by observed changes in climate (Fig. 1, second arrow). In the WGII contribution to the IPCC-AR5, an entire chapter was dedicated to the topic including the following definition: An impact of climate change is 'detected' if the observed state of the system differs from a counterfactual baseline that characterizes the system's behavior in the absence of changes in climate (IPCC 2014, chap. 18.2.1) and 'attribution' is the quantification of the contribution of climate change to the observed change in the natural, human or managed system. In both cases "climate change refers to any long-term trend in climate, irrespective of its cause" (IPCC 2014, chap. 18.2.1)."

In such set-up 'observed climate change' is defined as the specific realized long-term trend in climate captured by the central estimates of the model parameters. Therefore the parameters are not varied to capture the uncertainties of the 'true' parameters given potential other realisations of the historical data. In this regard the attribution of the magnitude of an extreme impact event to the 'realized' trend in climate is similar to the quantification of the 'Contribution of Observed Trend to Event Magnitude' discussed by (Diffenbaugh et al. 2017).

To highlight the difference to the probabilistic approach used climate attribution to anthropogenic forcing we have added the following paragraph to the main text, see lines 69 to 76:

"While in principle changes in natural, human and managed systems could also be attributed to anthropogenic climate forcing ('impact attribution to anthropogenic climate forcing', first and second arrow in Fig. 1, (Pall et al. 2011; Schaller et al. 2016; Mitchell et al. 2016)), we focus on 'impact attribution to climate change' as described in the WGII definition and introduce a climate dataset that can be used as input to climate impact models to characterize the system's behavior in the absence of climate change. As the response of natural, human or managed systems to climate and socio-economic forcings is commonly considered deterministic (or at least simulated in this way) the attribution of impacts to the observed realisation of climate change does not necessarily have to be addressed in a probabilistic way and gives more weight

to the separation of climate change from direct human influences as potential drivers of changes in the considered systems."

> 2) *Many authors have argued that impacts are felt in the extremes of climate more so than elsewhere, that is why counterfactual attempts are often made with very large model simulation sizes. This is not so easy in your methodology, although I can see ways forward for it – this should be discussed.*

Thank you very much for the comment. Indeed, we do not apply a probabilistic approach but rather extend the so called quantification of the 'Contribution of Observed Trend to Event Magnitude' discussed for weather extremes by Diffenbaugh et al. (2017) to the impacts of these weather extremes. This is now discussed in detail in the revised manuscript, see the revised introduction and lines 45 to 54:

"Climate attribution can refer to observed long-term trends (WGI contribution to IPCC 2013, chap. 10) or individual events (Trenberth, Fasullo, and Shepherd 2015; NAS 2016; Stott et al. 2016). Given the probabilistic setting, results are often formulated as statements such that 'Anthropogenic climate forcing has increased the probability of occurrence of the observed trend or the intensity or duration of a specific extreme event'. In a non-probabilistic framework the intensity of an observed event can be attributed to the observed realisation of climate change by comparing the event magnitude in the observed time series to the magnitude of the same event in a detrended version of the observed time series (quantification of the 'contribution of the observed trend to event magnitude', (Diffenbaugh et al. 2017)). This type of attribution to climate change does not address the reasons of the observed climate trend."

as well as lines 73 to 76:

"As the response of natural, human or managed systems to climate and socio-economic forcings is commonly considered deterministic (or at least simulated in this way) the attribution of impacts to the observed realisation of climate change does not necessarily have to be addressed in a probabilistic way and gives more weight to the separation of climate change from direct human influences as potential drivers of changes in the considered systems."

*The authors have worked in IPCC WG2 for a long time, and maybe a bit in WG1, but they need to be aware that their readers might be solely in one WG (or none at all), so the subject specific language needs to be very simple for a paper like this.*

We thank the reviewer for the helpful comment and hope that we have managed to adjust the language accordingly.

**Other corrections**

- *Title "counterfactual climate for impact attribution" – I see why you have this title, ´ but it seems that your work would be very useful*

We could not fully grasp this comment, and therefore have no proper reply.

- *Line 9: "anthropogenic" is needed ´ before climate change*

In most general terms, attribution is about the quantification of drivers of change. Impact attribution does not necessarily trace changes in impacts back to the anthropogenic causes of climate change. We rewrote the abstract to make our concept clearer.

- *Line 9: "Other drivers change according to observations". ´ Actually I think the other drivers should remain the same according to observations.*

This may be a misunderstanding. We aim to address problems with time-evolving impact drivers other than climate. They thus follow the observations, but still change. We rewrote the abstract to prevent this confusion.

- *Line 19-21: This sentence is very confusing without reading the paper, I suggest ´ making it stand alone.*

We rewrote the sentence and hope it is clearer now.

- *Line 26: Citation needed. Haustien et al, 2017 is a good ´ one, but there are others.*

We now cite the IPCC 1.5°C report, which has this statement in the Summary for Policymakers.

- *Figure 1: In this figure you show "climate change" as the affected quantity – this should just be "climate". Likewise for the driver in the second panel. I also do not agree with the caption that this is how the IPCC frame attribution.*

We updated the figure accordingly, with the aim to also clarify the link between attribution approaches and the potential to attribute impacts to emissions. We adapted the caption to be less prescriptive on the IPCC framing.

- *Lines 31-38: I like this description, and it is now clearer in my head ´ what you are doing. If this section can be summed up on 1 line for the abstract, that would really help make things clear from the start.*

Thank you for this remark! We now provide the description at the beginning of the abstract:

'Attribution in its general definition aims to quantify drivers of change in a system. According to IPCC WGII a change in a natural, human or managed system is attributed to climate change by quantifying the difference between the observed state of the system and a counterfactual baseline that characterizes the system's behavior in the absence of climate change, where "climate change refers to any long-term trend in climate, irrespective of its cause".'

- *Line 63-66: This section is ´ confusing me, in much the same way the end of the abstract did. Specifically, you say impact attribution does not need to address the causes of climate change. So, what is it addressing? You could state that explicitly here. I also think the attribution community would see this differently, and there is a danger that people will now be confused over what this term means.*

This is a key point to understand the presented concept, and we thank the reviewer for pointing this out. We now make this clearer at several points in the manuscript, including the abstract:

"Attribution in its general definition aims to quantify drivers of change in a system. According to IPCC WGII a change in a natural, human or managed system is attributed to climate change by quantifying the difference between the observed state of the system and a counterfactual baseline that characterizes the system's behavior in the absence of climate change, where "climate change refers to any long-term trend in climate, irrespective of its cause". … Attribution of climate impacts to anthropogenic forcing would need an additional step separating anthropogenic climate forcing from other sources of climate trends, which is not covered by our method."

See also lines 61 to 73 of the introduction:

"In addition to 'climate attribution', research on 'impact attribution' addresses the question: To what degree are observed changes in natural, human and managed systems induced by observed changes in climate (Fig. 1, second arrow). In the WGII contribution to the IPCC-AR5, an entire chapter was dedicated to the topic including the following definition: An impact of climate change is 'detected' if the observed state of the system differs from a counterfactual baseline that characterizes the system's behavior in the absence of changes in climate (IPCC 2014, chap. 18.2.1) and 'attribution' is the quantification of the contribution of climate change to the observed change in the natural, human or managed system. In both cases "climate change refers to any long-term trend in climate, irrespective of its cause" (IPCC 2014, chap. 18.2.1).

While in principle changes in natural, human and managed systems could also be attributed to anthropogenic climate forcing ('impact attribution to anthropogenic climate forcing', first and second arrow in Fig. 1, (Pall et al. 2011; Schaller et al. 2016; Mitchell et al. 2016)), we focus on 'impact attribution to climate change' as described in the WGII definition and introduce a climate dataset that can be used as input to climate impact models to characterize the system's behavior in the absence of climate change. "

- *Data section: What is the spatial scale of the dynamically ´ downscaled data? How much do we trust this data, especially in poorly observed parts of the world? What are the implications for these problems on the questions posed?*

The dynamical downscaling was done to a target resolution of approximately 0.5°. We expanded the data description section so that caveats are now better traceable. Concerning attribution studies using our data we now propose i) a series of control plots, ii) suggest to consider additional pairs of factual-counterfactual climate forcing data to better capture the

uncertainty induced by different climate forcing data sets and iii) suggest to always provide a detailed assessment of the explanatory power of the factual impact model simulations to overall reduce the risk of erroneous impact attribution induced by deficits in the observational data.

- *Line 101: Lots of work has been done on pattern scaling recently, so I think a more modern approach should be cited here. E.g. Herger, 2015, although many others exist as well.*

This is indeed true. We added the citation with a short explanation, see lines 242 to 244.

- *Line 101-105: This paragraph makes it much clearer in my head what you are doing. I would use some of this text to explain this earlier, especially about the non-causal link with global temperature.*

This is a good hint. We now state in the abstract that our approach aims at attribution to climate change, irrespective of the cause of climate change. We state in the introduction, see lines 78 to 80:

"The method proposed here is designed to generate a stationary climate without long-term changes. The statistical model used to produce this counterfactual climate removes the long-term change correlated with (but not necessarily caused by) large scale climate change, represented by GMT change instead of a simple temporal trend (see Methods). "

- *Page 7: Why are the distribution names bolded?*

We removed the bold font.

- *Line 144: I understand why you are including hurs, and commend it, but it is a bounded quantity (i.e. nominally constrained between 0-100), so would that cause problems when modelling with a Gaussian?*

We reworked the model for relative humidity (hurs) and now use a Beta distribution, which better represents the nature of this variable.

- *Figures 4 and 5: These are very nice and informative figures.*

Thanks for this positive judgement! They are now Figures 5 and 6.

- *Lines 255-256: I agree it is rare, but there are still numerous studies that have done this (see major comment).*

We now discuss these studies in the introduction section and set them in context to our work. See response to major comment.

- *Line 277-279: You should state clearly here why your data is useful in a complementary way to Gillett et al*

We now set our work in context to Gillett et al in the introduction, see lines 42 to 54. We also rewrote the section in the discussion section to clarify the complementarity of our dataset, see lines 470 to 484.

**References**

Challinor, A. J., J. Watson, D. B. Lobell, S. M. Howden, D. R. Smith, and N. Chhetri. 2014. "A Meta-Analysis of Crop Yield under Climate Change and Adaptation." *Nature Climate Change* 4 (4): 287–91.

Chen, Liang, and Oliver W. Frauenfeld. 2014. "A Comprehensive Evaluation of Precipitation Simulations over China Based on CMIP5 Multimodel Ensemble Projections: CMIP5 PRECIPITATION IN CHINA." *Journal of Geophysical Research* 119 (10): 5767–86.

Diffenbaugh, Noah S., Deepti Singh, Justin S. Mankin, Daniel E. Horton, Daniel L. Swain, Danielle Touma, Allison Charland, et al. 2017. "Quantifying the Influence of Global Warming on Unprecedented Extreme Climate Events." *Proceedings of the National Academy of Sciences of the United States of America* 114 (19): 4881–86.

Ghil, Michael, M. R. Allen, M. D. Dettinger, K. Ide, D. Kondrashov, M. E. Mann, Andrew W. Robertson, et al. 2002. "Advanced Spectral Methods for Climatic Time Series." *Reviews of Geophysics* 40 (1): 3–1.

Ghil, M., and R. Vautard. 1991. "Interdecadal Oscillations and the Warming Trend in Global Temperature Time Series." *Nature* 350 (6316): 324–27.

Giorgi, Filippo, and Raquel Francisco. 2000. "Evaluating Uncertainties in the Prediction of Regional Climate Change." *Geophysical Research Letters* 27 (9): 1295–98.

IPCC. 2013. *Climate Change 2013: The Physical Science Basis: Working Group I Contribution to the Fifth Assessment Report of the Intergovernmental Panel on Climate Change*. Cambridge University Press.

———. 2014. *Climate Change 2014 – Impacts, Adaptation and Vulnerability: Global and Sectoral Aspects*. Cambridge University Press.

Lange, Stefan. 2019. "Trend-Preserving Bias Adjustment and Statistical Downscaling with ISIMIP3BASD (v1.0)." *Geoscientific Model Development* 12 (7): 3055–70.

Li, Delei, Jianlong Feng, Zhenhua Xu, Baoshu Yin, Hongyuan Shi, and Jifeng Qi. 2019. "Statistical Bias Correction for Simulated Wind Speeds over CORDEX-East Asia." *Earth and Space Science (Hoboken, N.J.)* 6 (2): 200–211.

Lobell, David B., Wolfram Schlenker, and Justin Costa-Roberts. 2011. "Climate Trends and Global Crop Production since 1980." *Science* 333 (6042): 616–20.

Minoli, Sara, Christoph Müller, Joshua Elliott, Alex C. Ruane, Jonas Jägermeyr, Florian Zabel, Marie Dury, et al. 2019. "Global Response Patterns of Major Rainfed Crops to Adaptation by Maintaining Current Growing Periods and Irrigation." *Earth's Future* 7 (12): 1464–80.

Mitchell, Daniel, Clare Heaviside, Sotiris Vardoulakis, Chris Huntingford, Giacomo Masato, Benoit P. Guillod, Peter Frumhoff, Andy Bowery, David Wallom, and Myles Allen. 2016. "Attributing Human Mortality during Extreme Heat Waves to Anthropogenic Climate Change." *Environmental Research Letters: ERL [Web Site]* 11 (7): 074006.

NAS. 2016. *Attribution of Extreme Weather Events in the Context of Climate Change*. Washington, DC: The National Academies Press.

Pall, Pardeep, Tolu Aina, Dáithí A. Stone, Peter A. Stott, Toru Nozawa, Arno G. J. Hilberts, Dag Lohmann, and Myles R. Allen. 2011. "Anthropogenic Greenhouse Gas Contribution to Flood Risk in England and Wales in Autumn 2000." *Nature* 470 (7334): 382–85.

Rahmstorf, Stefan. 2007. "A Semi-Empirical Approach to Projecting Future Sea-Level Rise." *Science* 315 (5810): 368–70.

Schaller, Nathalie, Alison L. Kay, Rob Lamb, Neil R. Massey, Geert Jan van Oldenborgh,

Friederike E. L. Otto, Sarah N. Sparrow, et al. 2016. "Human Influence on Climate in the 2014 Southern England Winter Floods and Their Impacts." *Nature Climate Change* 6 (6): 627–34.

Schlesinger, Michael E., and Navin Ramankutty. 1994. "An Oscillation in the Global Climate System of Period 65–70 Years." *Nature* 367 (6465): 723–26.

Sheffield, Justin, Gopi Goteti, and Eric F. Wood. 2006. "Development of a 50-Year High-Resolution Global Dataset of Meteorological Forcings for Land Surface Modeling." *Journal of Climate* 19 (13): 3088–3111.

Stott, Peter A., Nikolaos Christidis, Friederike E. L. Otto, Ying Sun, Jean-Paul Vanderlinden, Geert Jan van Oldenborgh, Robert Vautard, et al. 2016. "Attribution of Extreme Weather and Climate-Related Events." *Wiley Interdisciplinary Reviews. Climate Change* 7 (1): 23–41.

Trenberth, Kevin E., John T. Fasullo, and Theodore G. Shepherd. 2015. "Attribution of Climate Extreme Events." *Nature Climate Change*, June 22, 2015.

Veldkamp, T. I. E., Y. Wada, J. C. J. H. Aerts, P. Döll, S. N. Gosling, J. Liu, Y. Masaki, et al. 2017. "Water Scarcity Hotspots Travel Downstream due to Human Interventions in the 20th and 21st Century." *Nature Communications* 8 (June): 15697.

Veldkamp, T. I. E., F. Zhao, P. J. Ward, H. de Moel, J C J, H. Müller Schmied, F. T. Portmann, et al. 2018. "Human Impact Parameterizations in Global Hydrological Models Improve Estimates of Monthly Discharges and Hydrological Extremes: A Multi-Model Validation Study." *Environmental Research Letters* 13 (5): 055008.

Wang, Xiaolan L., Y. Feng, G. P. Compo, V. R. Swail, F. W. Zwiers, R. J. Allan, and P. D. Sardeshmukh. 2013. "Trends and Low Frequency Variability of Extra-Tropical Cyclone Activity in the Ensemble of Twentieth Century Reanalysis." *Climate Dynamics* 40 (11): 2775–2800.

Weedon, G. P., Gianpaolo Balsamo, Nicolas Bellouin, Sandra Gomes, Martin J. Best, and Pedro Viterbo. 2014. "The WFDEI Meteorological Forcing Data Set: WATCH Forcing Data Methodology Applied to ERA-Interim Reanalysis Data." *Water Resources Research* 50 (9): 7505–14.

Weedon, G. P., S. Gomes, P. Viterbo, W. J. Shuttleworth, E. Blyth, H. Österle, J. C. Adam, N. Bellouin, O. Boucher, and M. Best. 2011. "Creation of the WATCH Forcing Data and Its Use to Assess Global and Regional Reference Crop Evaporation over Land during the Twentieth Century." *Journal of Hydrometeorology* 12 (5): 823–48.

---

## Referee Report (RR1)

This is a greatly improved version of the paper that was originally submitted – thank for. In my view, is nearly ready for publication. The main thing that I think remains to be done is to add some additional caveats that draw attention to nuances of interpretation that users will need to be aware of when applying the method and the dataset.

The following, therefore, lists a few additional comments and suggestions for the authors to consider. The comments are referenced by line number.

11:        The meaning of "counterfactual" should be stated in the abstract so that readers perusing GMD abstracts will understand what the method described in the paper is intended to produce.

73-76:     I find this statement problematic on a couple of levels.

           First, ironically, the methods used to produce the counterfactual climate are Bayesian, indicating that the parameters relevant to constructing counterfactual climate scenarios are described probabilistically. Shouldn't the fact that the intent of a Bayesian treatment is to quantify uncertainty at each stage of an analysis and propagate it appropriately to the next stage signal that a probabilistic treatment of impacts attribution is also needed?

           More fundamentally, this suggests that the impacts do not feed back onto the climate – but often they do, both by affecting the evolution of the forcing and thus the forced response of the climate and by affecting its internal variability, and thus altering the climate forcing that the impacted system is experiencing. Perhaps this only happens locally, but it could also happen on a large scale with potentially large implications for the evolution of the forced component of climate change (e.g., carbon cycle feedback from climate impacts on forest ecosystems). The impacts themselves are also likely subject to their own sources of internal (not forced by climate) variation that in turn might, or might not, affect the climate that is doing the forcing via feedbacks. One could think, for example, of a forest that is being impacted by the climate variations that it experiences, but that is also being impacted by insect disturbances, where insect population dynamics have their own internal variation that might not be entirely determined by climate. For these reasons, I would think that the attribution of impacts to climate would, in general, need to be treated in a probabilistic way, just as attribution of climate change to external forcing needs to be treated in that way.

           I suggest, therefore, that a bit more work be done to carefully nuance this statement.

104-111:   I think a few caveats are needed here. An implicit assumption is made here that an impacts model that is calibrated for the factual climate will continue to work equally well for counterfactual climates. That's not something that is necessarily a given. Consider, for example, a hydrologic model that is deployed on the Elbe River basin. Most hydrologic models of the Elbe would be carefully calibrated using observed

hydrologic quantities (streamflow, water temperature, etc) and observed meteorological drivers (air temperature, precipitation, wind speed and direction, solar radiation, etc) prior to using the model for prediction, historical reconstruction or future scenario development. This tuning is specific to the observational period that is used for calibration – with the result that the tuned model might not perform robustly in a different climate (e.g., with greater winter snow storage in the drainage basin in a cooler climate, or perhaps with greatly diminished snow storage in a future warmer climate). The point is that impacts models, just as climate models, are not entirely process based (as stated on line 119).

317-318: There are placeholders for subscripts that presumably should be removed.

319-320: There are many black and orange dots, so should "dot" be plural? Some explanation of the small dots and the single larger dot of each colour would also be in order.

333: Say what is meant by "physical bounds".

349-351: I imagine that this process of randomly turning dry days into wet days will result in some physical inconsistency with other variables. Perhaps a few words drawing attention to that possibility would be in order.

372: Suggest using the IPCC AR6 regions, if possible.

491: It would be good to say something about what constitutes "attribution". A calculation of the difference in impacts between factual and counterfactual climates (assuming that impacts can be determined with similar levels of confidence in both climates) would be a start, but attribution – drawing a causal connection and quantifying the change due to that cause – presumably requires careful arguments to rule out other confounding causes. At minimum, I think would need to be convinced a) that the change calculated with the impacts model is a reliable estimate of the change in the real world, b) that observed changes (to the extent that there is data) agree with the model simulated changes and c) that this similarity is not inadvertently due to confounding factors that affect the observed world but are perhaps not taken into account in the factual and counterfactual data used to drive the impacts model.

---

## Author Response (AR2)

*Referee comments in italic*

Responses in blue

"Quotes from the revised manuscript as plain text"

Review – ATTRICI v1.1 by Mengel et al.

*This is a greatly improved version of the paper that was originally submitted – thank for. In my view, is nearly ready for publication. The main thing that I think remains to be done is to add some additional caveats that draw attention to nuances of interpretation that users will need to be aware of when applying the method and the dataset.*

Thank you for these positive words! We address all comments in detail below. We include in the revision an update of the factual datasets GSWP3 (from v0.5b to v1.09) and W5E5 (from v1.0 to v2.0) that correct several errors and extend the covered period to 1901-2019. For more details see here. We apologize for the additional time this data update took. All line numbers refer to the clean version (no shown tracked changes) of the newly submitted manuscript.

*The following, therefore, lists a few additional comments and suggestions for the authors to consider. The comments are referenced by line number.*

*11: The meaning of "counterfactual" should be stated in the abstract so that readers perusing GMD abstracts will understand what the method described in the paper is intended to produce.*

Response: Thank you. We added the following explanation (line 11):

"... counterfactual baseline, which characterizes the system behaviour in the hypothetical absence of climate change, …"

This statement is from Stott et al. (2013), which fed into the IPCC WG2 AR5 definition of attribution.

*73-76: I find this statement problematic on a couple of levels.*

*First, ironically, the methods used to produce the counterfactual climate are Bayesian, indicating that the parameters relevant to constructing counterfactual climate scenarios are described probabilistically. Shouldn't the fact that the intent of a Bayesian treatment is to quantify uncertainty at each stage of an analysis and propagate it appropriately to the next stage signal that a probabilistic treatment of impacts attribution is also needed?*

*More fundamentally, this suggests that the impacts do not feed back onto the climate – but often they do, both by affecting the evolution of the forcing and thus the forced response of the climate and by affecting its internal variability, and thus altering the climate forcing that the impacted system is experiencing. Perhaps this only happens locally, but it could also happen on a large scale with potentially large implications for the evolution of the forced component of climate change (e.g., carbon cycle feedback from climate impacts on forest ecosystems). The impacts themselves are also likely subject to their own sources of internal (not forced by*

*climate) variation that in turn might, or might not, affect the climate that is doing the forcing via feedbacks. One could think, for example, of a forest that is being impacted by the climate variations that it experiences, but that is also being impacted by insect disturbances, where insect population dynamics have their own internal variation that might not be entirely determined by climate. For these reasons, I would think that the attribution of impacts to climate would, in general, need to be treated in a probabilistic way, just as attribution of climate change to external forcing needs to be treated in that way.*

*I suggest, therefore, that a bit more work be done to carefully nuance this statement.*

Response:

The reviewer is correct that in reality a) impacts feed back to the climate evolution and could alter climate variability and b) impacted systems may have internal variability. The statement therefore indeed needs more nuance. Concerning a), handling climate as external forcing to impacts is one of the simplifications at the core of ISIMIP, which allows to run impact models stand-alone and not as part of larger and more expensive climate models. This approach is taken up in ATTRICI to make impact attribution feasible for as many impact modellers as possible. Handling climate as external forcing is analogous to classic climate attribution where human emissions (e.g. greenhouse gas concentrations by RCPs) are handled as external forcing and changes in climate do not feed back to them. Our statement was also fuzzy concerning b): the ATTRICI approach does not hinder impact modellers to include internal variability - for example through insect population dynamics - and to investigate the role of such variability versus the external climate forcing. We now write (lines 74-80):

"The dataset is derived from the observed realization of climate, excluding the analysis how climate variability could produce alternative realizations of factual or counterfactual climate. The attribution approach is thus deterministic and not probabilistic, focusing on the separation of climate change from direct human influences as potential drivers of changes in the impacted systems. Concerning the internal variability within impacted systems, impact models to date largely do not resolve such variability and model a deterministic response to external drivers. Our approach would however allow for probabilistic attribution to climate change once impact models resolve internal variability."

We also made small adjustments to Fig. 1 to make the difference between attribution to climate change and attribution to anthropogenic forcing clearer.

Concerning the Bayesian approach, we use this to fit our model to the factual climate data only for numerical stability reasons, see lines 303-305. In fact, the posterior distributions we obtain for our model parameters are very narrow. They are too narrow to represent the true uncertainty of our counterfactual climate data because those posterior distributions neither represent the fundamental limitations of our detrending approach nor the data problems we discuss. To avoid creating a false impression of certainty, those Baysian uncertainties are not propagated. Instead, we advise users to apply our detrending approach to multiple factual datasets in order to at least quantify the climate input data-related uncertainty (lines 136-138).

*104-111: I think a few caveats are needed here. An implicit assumption is made here that an impacts model that is calibrated for the factual climate will continue to work equally well for counterfactual climates. That's not something that is necessarily a given. Consider, for example, a hydrologic model that is deployed on the Elbe River basin. Most hydrologic models of the Elbe would be carefully calibrated using observed hydrologic quantities (streamflow, water temperature, etc) and observed meteorological drivers (air temperature, precipitation, wind speed and direction, solar radiation, etc) prior to using the model for prediction, historical reconstruction or future scenario development. This tuning is specific to the observational period*

*that is used for calibration – with the result that the tuned model might not perform robustly in a different climate (e.g., with greater winter snow storage in the drainage basin in a cooler climate, or perhaps with greatly diminished snow storage in a future warmer climate). The point is that impacts models, just as climate models, are not entirely process based (as stated on line 119).*

Response: That is a valid point, thank you. We have added the following condition to the end of the paragraph (lines 114-115):

"This assumes that the climate impact model calibrated to perform well in the factual simulation performs robustly also with counterfactual climate input data."

*317-318: There are placeholders for subscripts that presumably should be removed.*

Response: Thanks for spotting this, we removed them.

*319-320: There are many black and orange dots, so should "dot" be plural? Some explanation of the small dots and the single larger dot of each colour would also be in order.*

Response: The larger dots are to illustrate one particular example of quantile mapping of a daily temperature value. To make this clearer we have slightly adjusted and extended the explanation in the text (lines 321-326):

"The counterfactual daily data are generated by quantile mapping, i.e. an observed value x that corresponds to a certain quantile of the factual distribution $A(T,t)$ is mapped to the counterfactual value x' that corresponds to the same quantile of the counterfactual distribution $A(T=0,t)$. We illustrate this for an observed value x that corresponds to the 95th percentile of the factual distribution in Fig. 4: We first obtain the cumulative probability of the factual (i.e. observed) temperature (large black dot in panel a) from the factual cumulative distribution function (CDF) (black line in panel b). We then obtain the counterfactual temperature (large orange dot in panel a) from the counterfactual CDF (orange line in panel b)."

We also made the figure caption clearer (Fig. 4):

"The single large black and orange point on the dashed vertical line in panel (a) highlight the factual and counterfactual value on October 25th."

*333: Say what is meant by "physical bounds".*

Response: Thank you for that. We are more specific now, writing (lines 338-339)

"We use the Gaussian distribution to model these variables as their values are far from their physical lower bound of zero."

*349-351: I imagine that this process of randomly turning dry days into wet days will result in some physical inconsistency with other variables. Perhaps a few words drawing attention to that possibility would be in order.*

Response: That is a valid point, thank you. We have extended this paragraph as follows (lines 357-358).

"This random conversion of dry days into wet days may result in physical inconsistencies with other climate variables. These inconsistencies are small by design since the new wet days are the least wet of all counterfactual wet days."

*372: Suggest using the IPCC AR6 regions, if possible.*

Response: This is a good suggestion, and we should indeed have used AR6 regions from the very start. As it would at this stage need a new analysis and the rewriting of the discussion of example regions, we would like to stick to the current definitions.

*491: It would be good to say something about what constitutes "attribution". A calculation of the difference in impacts between factual and counterfactual climates (assuming that impacts can be determined with similar levels of confidence in both climates) would be a start, but attribution – drawing a causal connection and quantifying the change due to that cause – presumably requires careful arguments to rule out other confounding causes. At minimum, I think would need to be convinced a) that the change calculated with the impacts model is a reliable estimate of the change in the real world, b) that observed changes (to the extent that there is data) agree with the model simulated changes and c) that this similarity is not inadvertently due to confounding factors that affect the observed world but are perhaps not taken into account in the factual and counterfactual data used to drive the impacts model.*

Response:

Thank you for this. The idea is in the manuscript, see for example line 107-110:

*"In a first step the climate impact model forced by observed climate and socio-economic drivers has to demonstrate to be able to reproduce the observed changes in natural, human and managed systems as measured by an impact indicator (comparison of black and blue solid lines in Figure 2). The attribution of the observed changes in natural, human and managed systems is built on a high explanatory power of the factual simulations."*

See also the second sentence in the Figure 2 caption:

*"First, in an evaluation step it has to be demonstrated that historical impact observations (black line) can be explained by the process-understanding as represented in the applied impact model and available knowledge about historical climate and socio-economic forcings."*

However, it deserves more clarity and the use of causality language. We therefore added a short paragraph to the discussion that builds on your ideas (lines 505-510):

"Attribution draws a causal connection and quantifies the change due to the cause. An important part of the attribution work is thus to ensure that the cause-effect relationship is correctly captured in the model. This requires careful analysis and model evaluation to show that the change estimated by an impact model is a reliable estimate of the real-world change. Simulated changes need to agree with observed changes and it needs to be ruled out that this agreement is due to confounding factors that drive observed change, but are not part of the model simulations. The ISIMIP3a historical simulations serve to address these points and demonstrate the explanatory power of impact models as an integral part of the attribution work."

**References**

Stone, D., Auffhammer, M., Carey, M., Hansen, G., Huggel, C., Cramer, W., ... & Yohe, G. (2013). The challenge to detect and attribute effects of climate change on human and natural systems. *Climatic Change*, *121*(2), 381-395.

---

## Author Response (AR3)

Dear Editor,

I updated the code and data availability section following your suggestions. I also requested an increased quota at zenodo, so that the data used in the manuscript is now fully archived at zenodo as well. I hope the updated code and data availability now meets the standards of GMD and I apologize again for my fuzzy handling of this earlier.

All the best and thanks,
Matthias Mengel on behalf of the author team